# Removing physiological motion from intravital and clinical functional imaging data

Sean C Warren[1,2]*, Max Nobis[1,2], Astrid Magenau[1,2], Yousuf H Mohammed[3], David Herrmann[1,2], Imogen Moran[2,4], Claire Vennin[1,2], James RW Conway[1,2], Pauline Mélénec[1], Thomas R Cox[1,2], Yingxiao Wang[5], Jennifer P Morton[6], Heidi CE Welch[7], Douglas Strathdee[6], Kurt I Anderson[6,8], Tri Giang Phan[2,4], Michael S Roberts[3,9], Paul Timpson[1,2]*

[1]Kinghorn Cancer Centre, Garvan Institute of Medical Research, University of New South Wales, Sydney, Australia; [2]St Vincent's Clinical School, Faculty of Medicine, University of New South Wales, Sydney, Australia; [3]Therapeutics Research Centre, Diamantina Institute, Faculty of Medicine, University of Queensland, Woolloongabba, Australia; [4]Immunology Division, Garvan Institute of Medical Research, Sydney, Australia; [5]Department of Bioengineering, Institute of Engineering in Medicine, University of California, San Diego, San Diego, United States; [6]Cancer Research UK Beatson Institute, Glasgow, United Kingdom; [7]Signalling Programme, Babraham Institute, Cambridge, United Kingdom; [8]Francis Crick Institute, London, United Kingdom; [9]Therapeutics Research Centre, School of Pharmacy and Medical Sciences, University of South Australia, Adelaide, Australia

**Abstract** Intravital microscopy can provide unique insights into the function of biological processes in a native context. However, physiological motion caused by peristalsis, respiration and the heartbeat can present a significant challenge, particularly for functional readouts such as fluorescence lifetime imaging (FLIM), which require longer acquisition times to obtain a quantitative readout. Here, we present and benchmark *Galene*, a versatile multi-platform software tool for image-based correction of sample motion blurring in both time resolved and conventional laser scanning fluorescence microscopy data in two and three dimensions. We show that *Galene* is able to resolve intravital FLIM-FRET images of intra-abdominal organs in murine models and NADH autofluorescence of human dermal tissue imaging subject to a wide range of physiological motions. Thus, *Galene* can enable FLIM imaging in situations where a stable imaging platform is not always possible and rescue previously discarded quantitative imaging data.
DOI: https://doi.org/10.7554/eLife.35800.001

*For correspondence:
s.warren@garvan.org.au (SCW);
p.timpson@garvan.org.au (PT)

Competing interests: The authors declare that no competing interests exist.

## Introduction

In recent years, a number of fluorescence imaging techniques such as fluorescence lifetime imaging microscopy (FLIM) have allowed researchers to visualize not only the structure but also the activity and function of molecules in living cells and tissues. Genetically-expressed Förster resonant energy transfer (FRET)-based biosensors enable researchers to probe signalling events in native tissues (*Conway et al., 2017*; *Ellenbroek and van Rheenen, 2014*; *Nobis et al., 2018*) where they can provide spatio-temporal information about drug target response in a tumour (*Conway et al., 2018*, *2014*; *Hirata et al., 2015*; *Nobis et al., 2017*, *2013*) or dynamic signalling events in migrating cells (*Mizuno et al., 2016*). Time resolved imaging of NADH autofluorescence (*Blacker and Duchen,*

**eLife digest** Understanding how molecules and cells behave in living animals can give researchers key insights into what goes wrong in diseases such as cancer, and how well potential treatments for these diseases work. A number of tools help us to see these processes. For example, fluorescent 'biosensors' change colour to tell us how active a particular protein is. This can indicate how well a drug works in different parts of a tumour.

High resolution microscopy makes it possible to image events happening in single cells, or even specific parts of a cell. However, small movements like those due to the heartbeat or breathing can blur the images, making it difficult to study living animals. This is particularly problematic for images that take several minutes to capture.

Warren et al. have now developed a new open source software tool called Galene. The tool can correct for small movements in images collected by a technique called fluorescence lifetime imaging microscopy (FLIM). As a result, clear images can be captured in situations that were not previously possible. For example, Warren et al. watched cancer cells migrating to the liver of a mouse from the spleen over 24 hours, and, using a fluorescent biosensor, showed that a repurposed drug interferes with how well the cells can attach to the liver. In addition, Warren et al. used the software to take steady 3D images of human skin in a volunteer's arm, which could be used to study drug penetration.

Galene could help researchers to study a wide range of biological processes in living animals. The software can also be applied to existing data to clean up blurred images. In the future Galene could be further developed to work with the imaging techniques used during surgery. For example, surgeons could use it to help them find the edges of tumours.
DOI: https://doi.org/10.7554/eLife.35800.002

*2016*; *Skala et al., 2007*) can be used to probe the metabolic state of cells and multispectral imaging (*Patalay et al., 2012*) has been investigated for the diagnosis of dermatitis and malignant melanoma (*König, 2012*), among other applications. Hyperspectral imaging, time resolved imaging in multiple spectral channels, can be used to extract microenvironmental information from autofluorescence (*Cutrale et al., 2017*). These techniques depend on measuring small changes in the fluorescence signal, such as a small change in lifetime or change in spectral properties. Consequently, more signal is required to determine the parameters of interest, often necessitating relatively long integration times. This requirement has proved to be a significant constraint to the uptake of FLIM in intravital microscopy, where physiological motion due to, for example peristalsis, respiration or the heartbeat can induce significant motion during the image acquisition. This motion can often be tolerated in intensity-based imaging where acquisition times are short. However, when an image must be integrated over tens or even hundreds of seconds, sample motion rapidly renders the image unintelligible. While physical restraints such as tissue clamping or the application of negative pressure, may be used to limit the sample motion to a degree, this approach is not always effective to the extent required for high resolution microscopy and, in some cases, can compromise the sample integrity. Given the increasingly wide application of both intravital and FLIM imaging, there is a growing need to enable the functional readouts provided by FLIM even in the presence of physiological motion (*Conway et al., 2014*). Here, we describe a motion blurring compensation approach using image-based realignment that can be applied directly to data acquired on existing commercial and clinical FLIM and conventional fluorescence microscopy systems in two and three dimensions in post processing.

FLIM is most commonly implemented using laser scanning microscopy (LSM). To acquire sufficient number of photons, an image is constructed by integrating the photon signal over many frames (passes over the scan area). A typical FLIM image may take several minutes to acquire and so is susceptible to motion blurring from physiological motion. Several image-based approaches have been used to correct for sample motion in intensity-based time lapse data without additional measurements in the context of LSM time series acquisitions. (i) Where the motion is slow relative to the frame rate, (e.g. a slow drift), rigid body image registration (*Thévenaz et al., 1998*) or feature-based registration such as Scale Invariant Feature Transform (SIFT)-based algorithms have been applied to

correct for the motion, (ii) When the motion is intermittent, frames captured during the motion may be automatically detected and excluded from the time series (*Soulet et al., 2013*), and (iii) When the motion is fast relative to the frame rate, each frame will appear distorted as the sample moves while the laser is scanned across the field of view. In this case, more advanced approaches that model the intra-frame motion using methods such as Hidden-Markov-Models (*Dombeck et al., 2007*), the Lucas–Kanade framework (*Greenberg and Kerr, 2009*), or algorithms based on Lie groups (*Vercauteren et al., 2006*) have been employed. These techniques have not, to date, been applied to FLIM data. This is because during FLIM acquisition, histograms of photon arrival times are typically accumulated to produce a single image. In this approach, blurring due to sample motion is 'baked into' the data and thus cannot be compensated. More recently, however, improved device-computer bandwidth and storage have enabled the recording of individual photon arrival times and markers associated with the scan frame and line clocks (*Becker et al., 2006*). Most modern commercial time-correlated single photon counting (TCSPC) FLIM systems support this mode, often by default.

Here, we describe an approach whereby we reconstruct each frame from this time-tagged FLIM data and determine the motion both between and within each frame using an approach based on the Lucas-Kanade framework (*Baker and Matthews, 2004*). We have implemented these algorithms in a new open source package, *Galene*, which can be used to correct for motion in two- and three-dimensional FLIM data collected using widely deployed commercial systems. We first evaluate the range of motions that can be effectively compensated using simulated data and compare the performance of *Galene's* core motion correction algorithms with open source and commercially available motion correction tools. We then validate our approach using intravital imaging of a number of FRET biosensors in vivo in a murine system and in clinical applications by imaging autofluorescence of human skin. While the main focus of this manuscript is motion correction of time resolved data, we show that *Galene* may also be used to correct conventional fluorescence microscopy data using intravital 3D multispectral imaging data of labelled immune cells in the murine lymph node highlighting its wider application for intravital imaging applications.

## Results

### Correction for motion in time-tagged, time-resolved FLIM data

The motion correction procedure is illustrated schematically in *Figure 1* and outlined in *Video 1*. We acquire FLIM data in a time-tagged time-resolved (TTTR) mode whereby each photon arrival time is recorded alongside frame and line markers, which allow us to locate each photon within the image. From these data, we first reconstruct the intensity of each constituent frame, or, in the 3D case, stack of frames (*Figure 1A–C*), that make up the image. We use these intensity data to determine the sample motion during the image acquisition relative to a reference stack (*Figure 1D–H*). For each stack, we first perform a fast, rigid realignment using a 3D generalisation of the phase correlation method (*Foroosh et al., 2002*). This corrects for coarse sample displacements but cannot correct for sample motion during the stack, which leads to distortions, rather than simply displacement of the stack. To estimate the displacement of the sample during the stack acquisition, we use a fitting approach where we account explicitly for the raster scan pattern used to acquire the data following the approach of *Greenberg and Kerr (2009)*. The microscope takes a finite time to scan over the stack. We assume that the sample moves linearly between a series of initially unknown two- or three-dimensional displacements spaced equally through the scan duration. For a given set of displacements, we know where the sample was when the microscope acquired each pixel, and we can thus reconstruct the undistorted stack by three-dimensional interpolation and compare this reconstruction to the reference stack. By estimating the motion at a number of points across the image, we can account for motion in both the fast-axis, which appears as a 'wave-like' pattern in the data, and in the slow axis, which appears as compression or expansion of sections of the image. We can then determine the sample motion during the stack acquisition by finding the set of displacements which minimises the difference between the corrected stack and the reference stack using a trust-region non-linear optimisation algorithm. This approach requires that we compute the Jacobian of this error function, that is the gradient of the difference between the two stacks at each pixel with respect to the unknown displacement parameters. To perform this optimisation efficiently we use a variant of

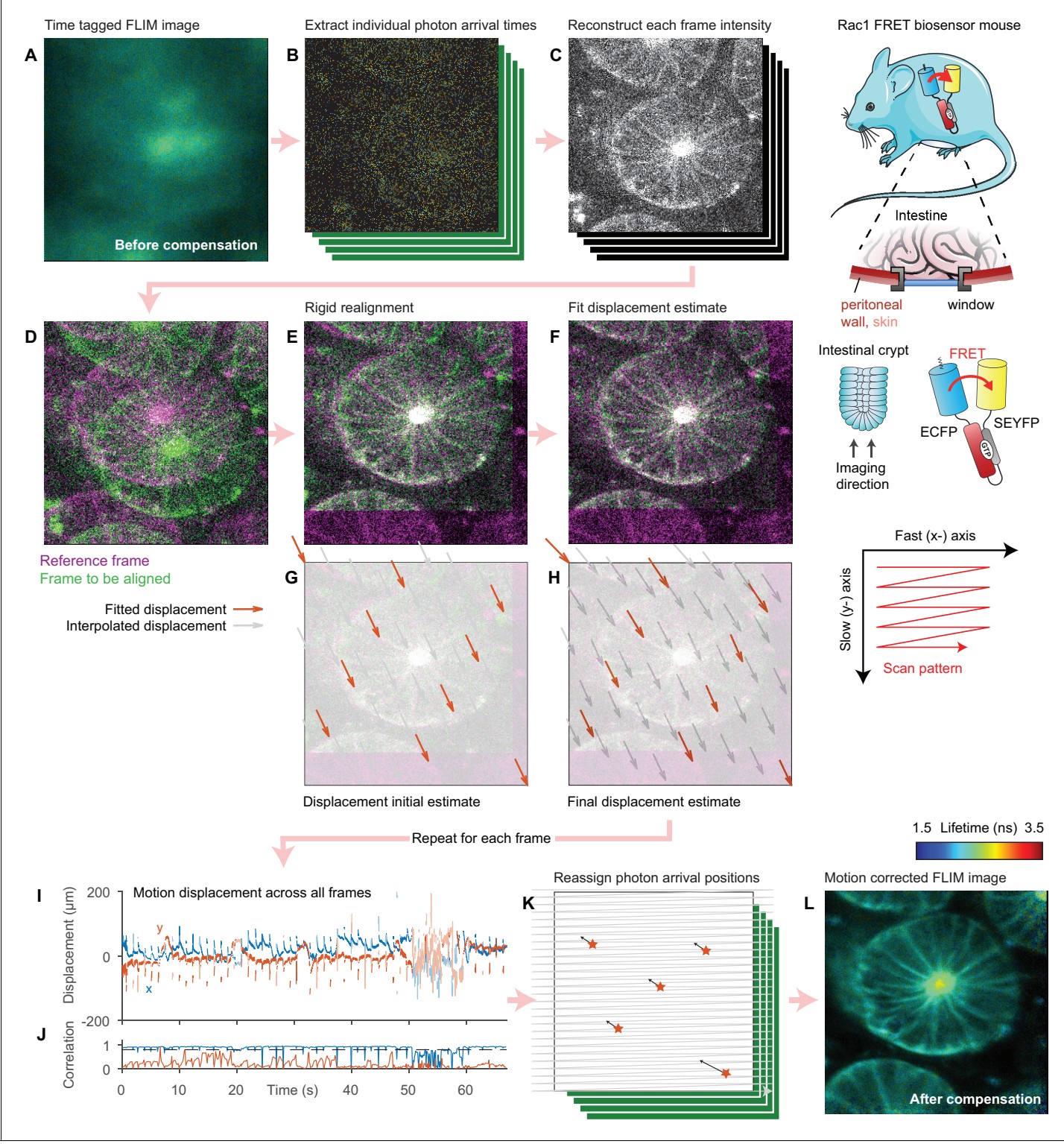

**Figure 1.** Illustration of motion correction procedure. (**A**) Intensity merged FLIM image of intestinal crypt acquired in vivo using a titanium optical window from a Rac1 FRET biosensor mouse, composed of 443 separate frames. (**B**) As the data are acquired in a time-tagged mode, the arrival time and position of each photon in the dataset can be extracted. (**C**) Using these data, the intensity of each frame can be reconstructed. (**D**) One frame is selected as the reference (magenta). To realign subsequent frames (example shown in green), (**E**) a rigid realignment step is first performed, estimating the coarse offset between the images. (**F**) To estimate the motion during the frame leading the residual difference between the two images, we select a number of points equally spaced in time during the frame scan and estimate the displacement of the sample at each point (red arrows). (**G**) We use the

*Figure 1 continued on next page*

*Figure 1 continued*

rigid realignment as the initial estimate for the displacement. (**H**) As the fit progresses, the displacement estimates reduce the difference between the two images. The displacement of pixels between each point are computed using linear interpolation (grey arrows). The (**I**) displacement during each frame and (**J**) correlation between each frame and the reference is computed by repeating this process for each frame. (**K**) The FLIM image is reprocessed and each photon arrival is reassigned to its estimated origin point on the sample using the interpolated displacements. (**L**) The final intensity merged FLIM image can then be analyzed using conventional approaches. Mouse and intestine illustrations were adapted from Servier Medial Art, licensed under the Creative Commons Attribution 3.0 Unported license.

DOI: https://doi.org/10.7554/eLife.35800.003

the inverse-compositional Lucas-Kanade algorithm (*Lucas and Kanade, 1981*), used extensively in image registration applications (*Baker and Matthews, 2004*); its key insight is that, rather than calculating the gradient of the interpolated stack at every iteration, a computationally expensive procedure, we can instead use the gradient of the reference stack, which is of course invariant (*Baker and Matthews, 2004*). This optimisation yields an estimate of the sample motion during the scan. With this sample displacement information, the FLIM image can be reconstructed accounting for the motion, reassigning each photon arrival to the correct pixel producing a distortion-free image (*Figure 1I–L*). We use the displacement information to determine the effective dwell time in each pixel (which will vary across the image due to the sample motion). This information is stored alongside the corrected FLIM data and may be used when displaying intensity-merged FLIM data.

## Evaluating performance limits using simulated FLIM data

To determine the range of amplitudes and frequencies of sample motion that can be reliably corrected, we used Monte Carlo simulations of FLIM data in the presence of sample motion (*Figure 2A*). We compare these results with reference to average values for physiological events known to impair image acquisition; the heartrate (red arrow, 350 bpm, ~6 Hz) and respiratory rate (blue arrow, 60 bpm, 1 Hz) of an adult mouse anaesthetized under ~1% isoflurane (*Ewald et al., 2011*) are shown. *Figure 2B* shows the average correlation coefficient, a measure of how well the correction has performed, for a range of frequencies and amplitudes of motion relative to the field of view (FOV) size. We observed that the system could compensate for motion parallel to the fast axis (0°) more effectively than motion parallel to the slow axis (90°); this is unsurprising as motion in the slow axis will result in whole lines of the sample being sampled either twice or not at all. *Figure 2C–G* illustrate several example alignment results with different motion conditions; the red dots indicate the location on the frequency-magnitude plot shown in *Figure 2B* for each condition. *Figure 2C and D* show, respectively, slow (1.5 Hz) and fast (6 Hz) motion with the same magnitude of ~10% of the FOV aligned with scanner fast axis. Here, we can effectively correct for the motion as shown in the realigned images and estimated displacements plotted with the simulated displacements. *Figure 2E* shows fast motion at 45° to the fast axis; again, we can effectively compensate for this motion although the resultant realigned image is marginally degraded. *Figure 2F* shows a larger (higher magnitude) motion, approximately ~20% of the field of view at 45° to the fast axis. In this case, we are not able to effectively compensate for the motion, and the realigned image is significantly degraded. We note that a similar motion along the fast axis could be corrected (see matching point on *Figure 2Bi*). *Figure 2G* shows a very large (~30% of the field of view) motion along the slow axis, which we are unable to correct. We note that the range of amplitudes and frequencies that *Galene* is able to correct covers a broad range of physiologically relevant motions commonly observed during functional intravital imaging and will therefore be of use in a range of in vivo imaging applications.

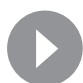

**Video 1.** Video abstract. Overview of motion correction using *Galene*.

DOI: https://doi.org/10.7554/eLife.35800.004

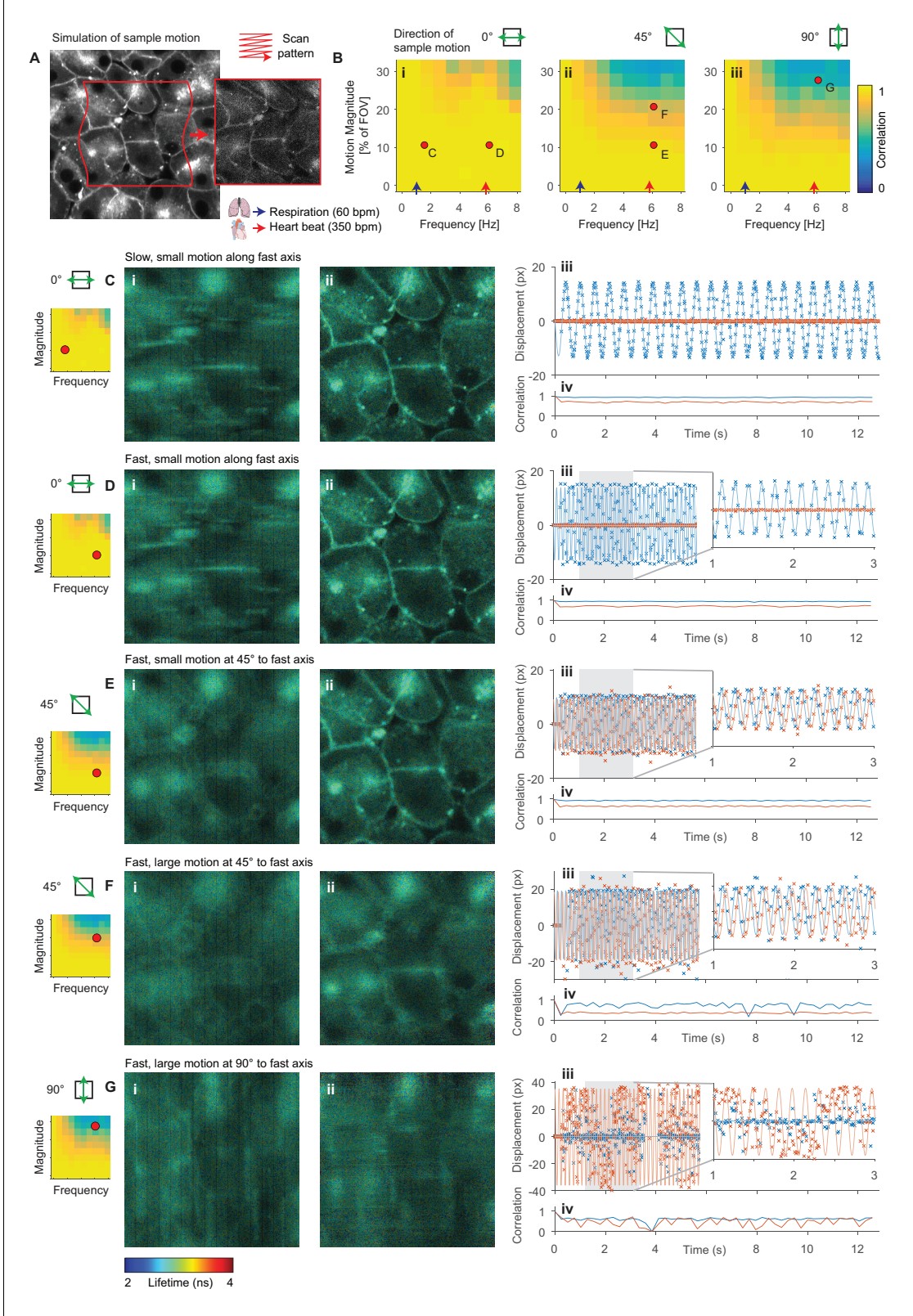

**Figure 2.** Evaluation of motion correction performance with simulated data. (**A**) Illustration of generation of a frame in a simulated TTTR dataset. Main image shows high SNR intensity image used as the reference sample intensity. Red lines indicate the sample motion during the illustrated frame. Inset image shows the resultant simulated frame. (**B**) Average correlation coefficient over a range of magnitudes and frequencies of sinusoidal sample motion along an axis (i) 0°, (ii) 45° and (iii) 90°, respectively, from the fast axis. For comparison, indicative average values for the heart rate (*red arrow*, 350 bpm,

*Figure 2 continued on next page*

*Figure 2 continued*

5.8 Hz) and respiration rate (*blue arrow*, 60 bpm, 1 Hz) of an adult mouse anaesthetized under ~1% isoflurane are shown. Black dots indicate results illustrated in the following panels. (**C–G**) Exemplar simulation results showing intensity merged lifetime images (i) without and (ii) with motion compensation. (iii) Estimated displacement traces in (*blue*) x and (*red*) y directions over time. In D-Giii, inset panels show expanded views of displacements between 1 and 3 s. (iv) Correlation between reference frame and (*red*) uncorrected and (*blue*) corrected images over time. Examples shown in C–E were corrected successfully, while the motion in examples F and G was too large to effectively compensate. (**C**) Motion parallel to fast axis with frequency 1.5 Hz and magnitude 10% of FOV. (**D**) Motion parallel to fast axis with frequency 6 Hz and magnitude 10% of FOV. (**E**) Motion at 45° to fast axis with frequency 6 Hz and magnitude 10% of FOV. (**F**) Motion at 45° to fast axis with frequency 6 Hz and magnitude 20% of FOV. (**G**) Motion at 90° to fast axis with frequency 6 Hz and magnitude 28% of FOV.

DOI: https://doi.org/10.7554/eLife.35800.005

The following figure supplement is available for figure 2:

**Figure supplement 1.** Benchmarking of intensity-only motion correction performance with simulated data.

DOI: https://doi.org/10.7554/eLife.35800.006

## Comparison with other software packages for intensity-based realignment

We compared the core motion correction algorithm used by *Galene* with three open source motion correction packages using intensity-only data. We used two ImageJ plugins, StackReg, implementing a rigid registration algorithm (*Thévenaz et al., 1998*) and 'Linear Stack Alignment with SIFT', an approach based on the Scale Invariant Feature Transform (*Lowe, 2004*). We also evaluated the python package SIMA (*Kaifosh et al., 2014*), which uses a Hidden Markov Model (HMM)-based approach (*Dombeck et al., 2007*). We generated simulated time lapse intensity data with sample motion at 45° to the fast axis over a range of amplitudes and frequencies. We performed motion correction of these data with each software package and plotted the average correlation between each motion corrected frame and the reference frame as a function of frequency and amplitude of motion (*Figure 2—figure supplement 1*). StackReg and SIFT both correct for rigid transformations between each frame and so are unable to cope with the distortion produced by LSM with a moving sample; consequently, these packages are only able to correct for low frequency motion (as, for example, encountered during slow sample drifts where the observed motion artefact can be approximated by a linear transformation (for example, *Figure 2—figure supplement 1B*). SIMA uses a HMM model which uses information about the laser scan pattern and so is able to cope with a larger range of motion (for example, *Figure 2—figure supplement 1C*, equivalent to a small displacement caused by the heartbeat). However, *Galene* is able to correct for significantly larger sample motions than SIMA (compare *Figure 2—figure supplement 1Aiv*, SIMA and v, *Galene*), for example the larger motion shown in *Figure 2—figure supplement 1D*, corresponding to a larger motion induced by the heartbeat, and so will be useful in a broader range of intravital experimental conditions.

## Evaluation of the effect of image scan configuration on motion correction performance

We went on to evaluate the use of *Galene* in an intravital imaging setting. As demonstrated using the simulated data, the speed and direction of the motion has a significant effect on the extent to which we are able to correct the data. The critical parameter is, in fact, the relative speed of the motion with respect to the scan rate; a 5 Hz motion acquired with a frame rate of 1 Hz will appear to oscillate five times during each frame acquisition while the same speed acquired with a frame rate of 5 Hz will only appear to oscillate once per frame. Since the scan rate is generally a user controllable parameter, we imaged the same region at different scan rates to determine how this parameter affects the motion correction performance.

We acquired images through surgically implanted titanium windows (see schematic in *Figure 3*) that enable longitudinal imaging of abdominal organs (*Ritsma et al., 2014*). Imaging abdominal organs through an optical window is challenging as they experience considerable motion as a result of physiological activity in nearby organs such as respiration and the heartbeat. Simply finding areas sufficiently stable to acquire FLIM images significantly limits the usable area of the window, and, in some cases, can render mice completely unusable. Using these windows, we imaged the pancreas in a genetically engineered mouse expressing a FRET biosensor for the small GTPase Rac1 (*Itoh et al., 2002*; *Johnsson et al., 2014*). We analyzed the FLIM data before and after motion correction by

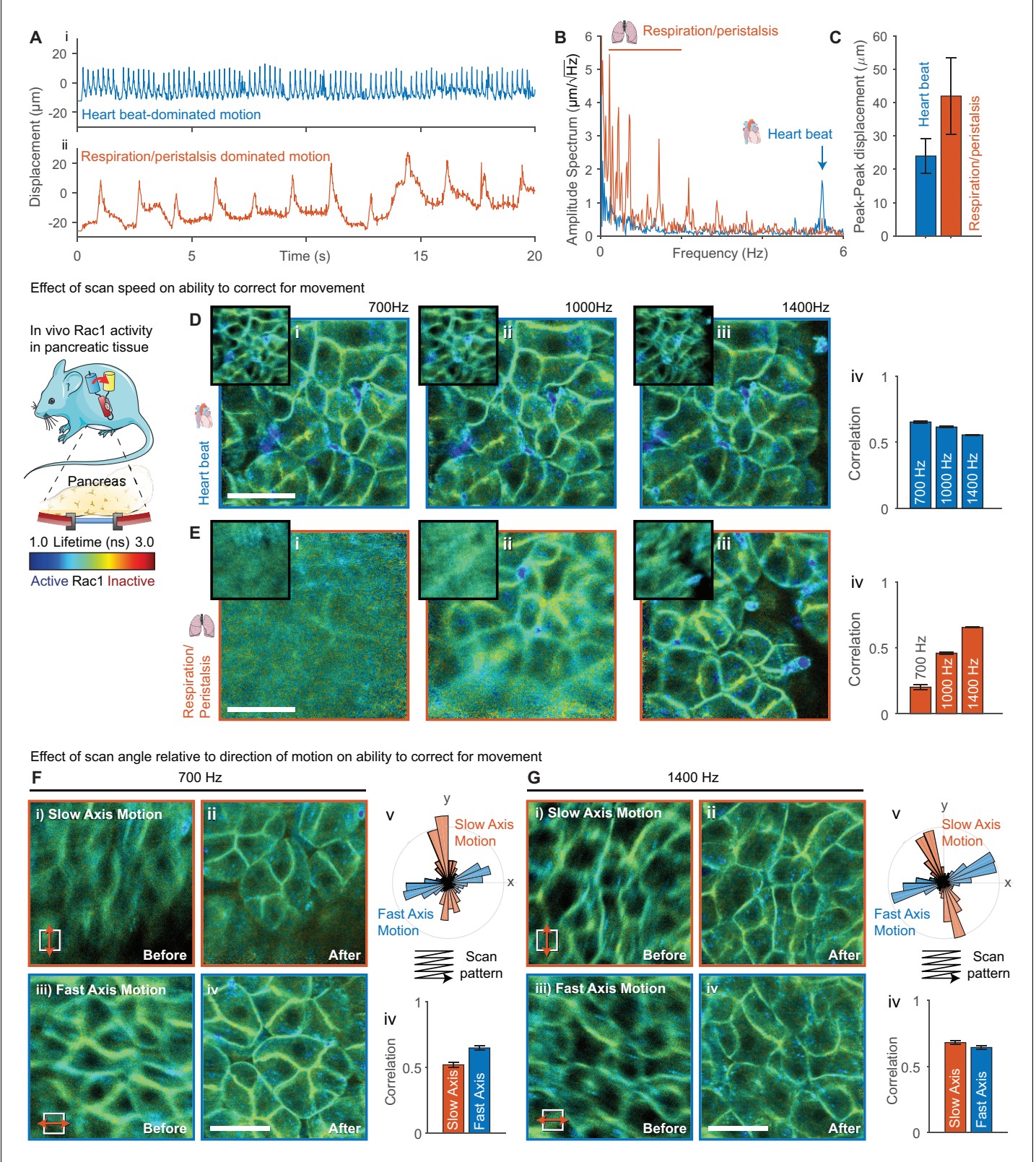

**Figure 3.** Evaluation of effect of scan speed and angle on motion correction performance. (**A**) Imaging of pancreatic tissue in vivo in a Rac1 FRET biosensor mouse using an abdominal titanium imaging window. (**A**) Estimated displacement for first 20 s of motion (i) dominated by heartbeat and (ii) dominated by respiration and peristalsis recorded at 1400 Hz. (**B**) Amplitude spectrum of displacements for two data series showing dominant contributions from the heartbeat (~5.6 Hz) and respiration and peristalsis (0.5–2.5 Hz). (**C**) Peak-to-peak displacements for the data series. Error bars

*Figure 3 continued*

show standard deviation over series. (D,E) Motion corrected FLIM images acquired in (D) heartbeat dominated regime and (E) respiration dominated regime at (i) 700 Hz, (ii) 1000 Hz and (iii) 1400 Hz, (iv) average correlation of corrected frames over series. (F,G) FLIM data were acquired with predominant motion occurring along the slow scan axis (*red, top-bottom*) and fast scan axis (*blue, left-right*) at (F) 700 Hz and (G) 1400 Hz. FLIM images are shown (i) before and (ii) after motion correction for motion along the slow axis and (iii) before and (iv) after motion correction for motion along the fast axis. (v) Angular histogram of displacements for data acquired along the slow and fast axis. (iv) Average correlation of data acquired with motion aligned along the slow and fast axis. White scale bars 100 μm. Mouse and pancreas illustrations were adapted from Servier Medial Art, licensed under the Creative Commons Attribution 3.0 Unported license.

DOI: https://doi.org/10.7554/eLife.35800.007

The following figure supplement is available for figure 3:

**Figure supplement 1.** Evaluating the effect of realignment parameters on motion correction performance.

DOI: https://doi.org/10.7554/eLife.35800.008

fitting each pixel to a single-exponential model as previously demonstrated (*Johnsson et al., 2014*). We acquired FLIM-FRET images in two locations with different motion patterns, (i) where motion was dominated by the heartbeat and (ii) where motion is dominated by respiration and peristalsis (see displacement traces shown in *Figure 3Ai,ii*) at 700, 1000 and 1400 Hz line rates. We performed an amplitude-spectrum analysis on the estimated displacements (*Figure 3B,C*) and found motion of (i) a frequency of 5.4 Hz and peak-peak amplitude of 27 μm (approximately 10% of the FOV), corresponding to the heartbeat and (ii) slower motions with a range of frequencies from 0.2 to 2.2 Hz with an average peak-peak amplitude of 42 μm (approximately 15% of the field of view), corresponding to respiration. We found that for the faster but smaller motion due to the heartbeat we were able to correct for motion equally well at all scan rates (see *Figure 3D*, average correlation quantified in *Figure 3Div*). Note that for the same correction performance the correlation is reduced slightly at higher scan rates due to the lower signal to noise in each image. For the larger due to respiration and peristalsis, we found that we were only able to successfully correct for motion at 1400 Hz scan rates; partial correction was obtained at 1000 Hz and effectively no correction at 700 Hz (see *Figure 3E*). Underscoring the simulated data results, these data indicate that for larger motions it may be helpful to acquire at a faster frame rate where possible.

We went on to evaluate the effect of the relative alignment between the dominant direction of motion and the fast scanner axis. We imaged the pancreas with the motion aligned with the slow axis (red) and fast axis (blue) at 700 and 1400 Hz (see *Figure 3F and G* respectively) by rotating the microscope scan field. *Figure 3F, Gv* shows an angular histogram of the displacements showing the alignment with the scanner axes for the two cases and *Figure 3F, Giv* shows the average correlation between the realigned frames. At 700 Hz the realignment is significantly improved when the motion is aligned with the fast axis. At 1400 Hz, when the motion is slower relative to the scan rate, the motion is corrected equally well in either case. This highlights both the importance of using a fast scan rate where possible and, where there is a clear direction of motion, approximately aligning it with the scanner fast axis by rotating the scanner field of view.

## Evaluation of the effect of realignment parameters on motion correction performance

There are a number of user controllable options for performing the realignment. The first is the number of realignment points used across each frame of the image. The motion is interpolated linearly between these points across the image. In principle, using a larger number of points allows correction of higher frequency motions. We acquired 256 × 256 images of the pancreas and realigned the data using either coarse translation-only information determined using phase correlation or realignment with 3, 5, 10, 20 and 40 points per image (*Figure 3A*, quantified in *Figure 3B*). For images of this size, there was a slight improvement in the average correlation with increasing number of realignment points up to 20 points. Using 40 points, however, the correlation was slightly reduced. Using a very large number of realignment points can be detrimental; as the number of points increases, the number of pixels which constrain each point is reduced and eventually there is not enough information to accurately determine the motion at each point. In general, we found that using between 5–20 realignment points for 256 × 256 images and 10–20 points for 512 × 512 images was sufficient to obtain good correction over a broad range of conditions.

As the signal to noise level in each frame of a FLIM image is often low due to the restricted count rate requirements of TCSPC imaging, a Gaussian smoothing kernel may optionally be applied to each frame before realignment to improve the realignment. This smoothing is only applied in the x (fast-) axis so that pixels are only convolved with those acquired immediately before and after. To evaluate the effect of the degree of smoothing, we realigned an image with low signal to noise with a range of smoothing kernel widths between 0 and 10 pixels (*Figure 3C*, quantified in *Figure 3D*). We normally use the correlation between the smoothed images to reduce the effect of noise on the realignment result quantification. Here, however, we use the correlation between the unsmoothed images to allow us to compare the correlation between different smoothing kernels. We found a noticeable improvement in the realignment result using a kernel of with 2 or 4 pixels compared to no smoothing. At larger kernel sizes, the correlation gradually reduced; using excessively large smoothing kernels can reduce the quality of the realignment as the contrast in the image is reduced. We have found that using a smoothing kernel of 2–4 pixels works well over a broad range of conditions.

## Imaging intestinal crypts in vivo and ex vivo in a Rac1 FRET biosensor mouse

We went on to use *Galene* to image Rac1 activity in the intestinal crypts. Rac1 regulates a diverse array of cellular events including the cell cycle, cell-cell adhesion, motility and differentiation (*Heasman and Ridley, 2008*) and has been shown to be a key driver of Wnt-induced stem cell activation within the intestinal crypt (*Myant et al., 2013*). The Rac1 biosensor contains an ECFP donor, which has a complex decay profile, dominated by contributions from two conformations with similar spectral profiles. Here, we have fitted the data to a complex-donor FRET model previously described (*Warren et al., 2013*) consisting of two contributions with different levels of FRET. Using global analysis, we determined the FRET efficiencies of the Rac1 GTP (active) and GDP bound (inactive) states to be $E = 0.65$ and $E = 0.02$. By fitting the contributions of each component, we can estimate the fraction of active Rac1 biosensor in each pixel as shown in *Figure 4A*.

Imaging of the intestine is further complicated by motion induced by peristalsis, wave-like contractions of the digestive tract which propel food through the intestine. The gut can be attached only gently to the window (*Ritsma et al., 2012*) as immobilizing a tract of the intestine can obstruct the bowel. The movement caused by peristalsis almost completely obscures the sample structure when imaging for even a few seconds (*Figure 4Ai*). *Video 2* shows the individual frames from the acquisition with and without correction and the accumulated time resolved image. When imaging crypts we noted that, alongside persistent smaller motion, occasional large, transient displacements occur where the crypt under observation moves completely out of the field of view (shown in the displacement estimates, *Figure 4A*iii); this, of course, cannot be compensated. We automatically identify and remove these frames by applying a threshold to the correlation between the reference image and the best estimate of the corrected frame, in this case 0.8 (*Figure 4Aiv*), discarding frames with lower correlation values. *Figure 4A* shows examples of frames which were successfully corrected (vi before correction, vii after correction) and frames which were excluded from the reconstructed image (viii, ix). Using this procedure, we can successfully recover an undistorted image of the live crypt (*Figure 4Aii*), and, by fitting to a complex-donor FRET model, estimate the fraction of the active biosensor in each pixel.

Peristaltic motion continues even ex vivo when the intestine is maintained appropriately for live cell imaging. We imaged intestinal crypts in freshly excised tissue from the Rac1 and observed significant peristaltic motion which we could effectively correct using *Galene*, as illustrated in *Figure 4B* and *Video 3*. To inhibit peristalsis for imaging, researchers often use scopolamine, a small molecule muscarinic antagonist that inhibits the contraction of the smooth muscle layer surrounding the intestine (*Wang et al., 2008*). To compare this approach to image based correction, we imaged tissue with and without pre-treatment with scopolamine. To quantify the data, we first used phasor analysis (*Figure 4C*, phasor analysis of image shown in *Figure 4B*) to separate the biosensor fluorescence (blue gate) from the tissue autofluorescence (red gate). We then manually segmented single cells and computed the average fraction of active Rac1 biosensor. Treatment with scopolamine effectively inhibited peristalsis; however, unexpectedly we also observed a significant activation of Rac1 in tissue treated with scopolamine compared to untreated tissue (*Figure 4D*, quantified in E) This activation was of a similar magnitude to that of tissue treated with phorbol myristate acetate (PMA)

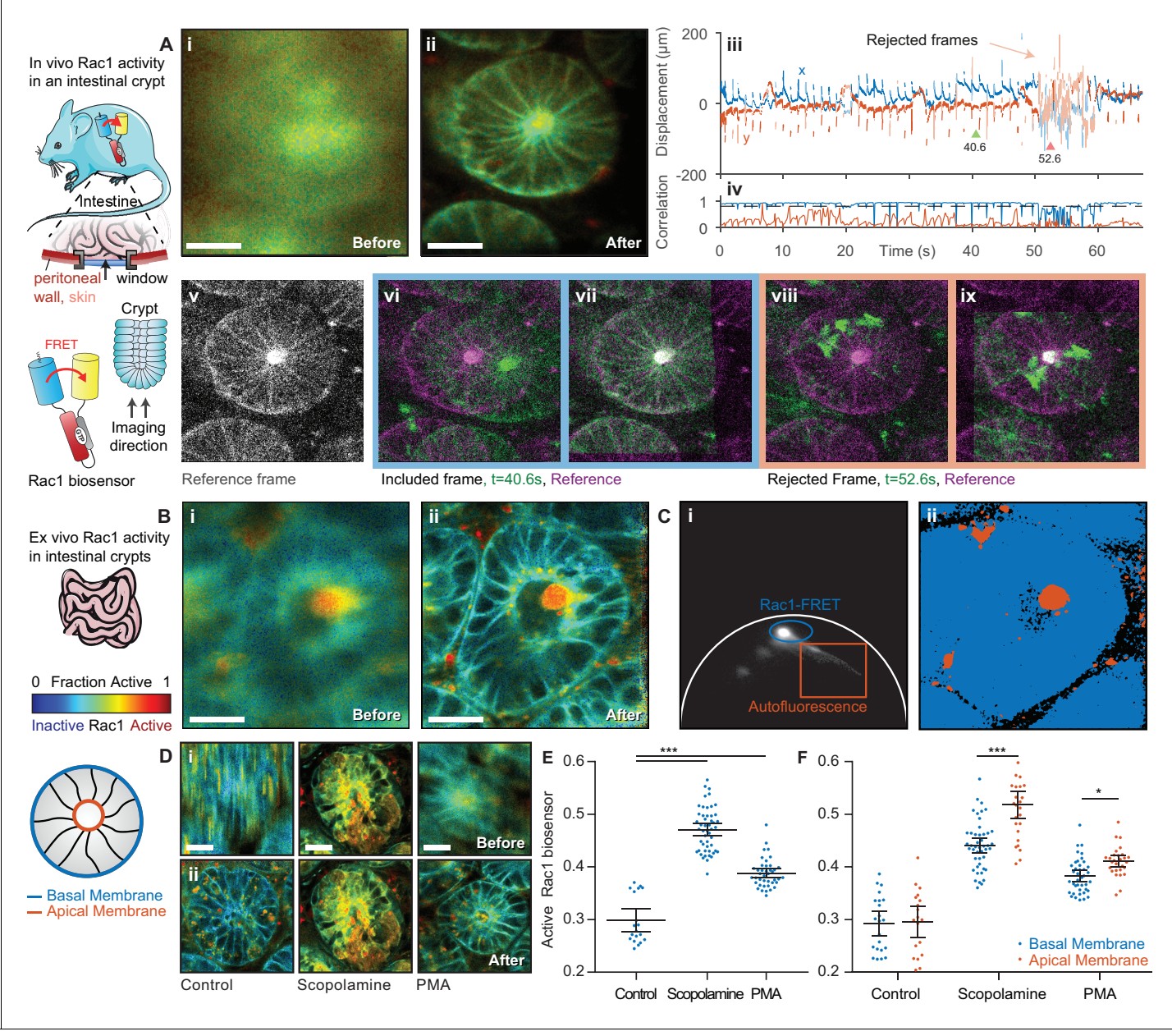

**Figure 4.** FLIM of intestinal crypts from the Rac1-FRET biosensor mouse in vivo and ex vivo using motion compensation. (**A**) Imaging of intestinal crypts in vivo using an abdominal titanium imaging window (i–ii) Example FRET biosensor activity maps (i) before and (ii) after motion compensation showing fraction of active FRET biosensor determined by fitting to a FRET model accounting for the complex exponential decay of ECFP. White scale bars, 50 μm. (iii) Estimated displacement traces in (*blue*) x and (*red*) y directions over time. Pastel shaded regions indicate frames that could not be successfully corrected with correlation coefficients < 0.8. (iv) Correlation between reference frame and (*red*) uncorrected and (*blue*) corrected images over time. Black dashed line denotes threshold (0.8) used to reject frames which could not be corrected. (**v**) Selected reference frame (vi,vii) Example of a successfully corrected frame (vi) before and (vii) after correction. (viii,ix) Example of a frame that could not be corrected. (**B**) Imaging of intestinal crypts ex vivo. (i–iv) as (**A**), no correlation threshold applied. (**C**) (i) Phasor plot of image shown in (**B**) to separate biosensor fluorescence (*blue*) from autofluorescence (*red*) and i) back projection of selected gates. (**D**) Intensity merged lifetime images of crypts (i) before and (ii) after motion compensation treated with (*left-right*) no drug, 200 nM PMA or 1 μM scopolamine. White scale bars 50 μm. (**E**) Quantification of fraction of active biosensor in crypts after drug treatment. (**F**) Subcellular analysis of fraction of active biosensor in basal (*blue*) and apical (*red*) membranes after drug treatment, shown per cell. Error bars show means ± SEM. **p<0.01; ***p<0.001 using one-way ANOVA. Mouse and intestine illustrations were adapted from Servier Medial Art, licensed under the Creative Commons Attribution 3.0 Unported license.

DOI: https://doi.org/10.7554/eLife.35800.009

The following source data and figure supplements are available for figure 4:

**Source data 1.** Source data for graphs show in *Figure 4E and F*.

*Figure 4 continued on next page*

*Figure 4 continued*

DOI: https://doi.org/10.7554/eLife.35800.012

**Source data 2.** Source data for graph show in *Figure 4—figure supplement 1C*, showing (Sheet 1) average optical densities for active-Rac1 IHC staining per mouse and (Sheet 2) individual optical densities for active Rac1 IHC staining per cell for each mouse.

DOI: https://doi.org/10.7554/eLife.35800.013

**Figure supplement 1.** IHC for Rac1-GTP in intestinal crypts.

DOI: https://doi.org/10.7554/eLife.35800.010

**Figure supplement 2.** Benchmarking of intensity-only motion correction performance with frames from intestinal crypts.

DOI: https://doi.org/10.7554/eLife.35800.011

(*Johnsson et al., 2014*), a potent small molecule activator of Rac1. This effect was also observed when fixed tissue was stained with a Rac1-GTP specific antibody (*Myant et al., 2013*) (representative images and quantification shown in *Figure 4—figure supplement 1*). This interference with Rac1 signalling highlights the need to ensure pharmacological approaches to reducing sample motion do not affect the process under observation. In contrast, by using image based correction, these artefacts can be avoided when looking, for example, at subtle changes in Rac1 GTPase regulation which is known to drive stem cell activity in intestinal crypts (*Myant et al., 2013*).

Without motion correction, it is extremely difficult to identify subcellular compartments in a majority of the intestinal crypt data. After correcting for motion, however, we are able to robustly identify subcellular regions and structures in the data. We performed sub-cellular analysis of Rac1 activity in the basal and apical membranes of intestinal crypts with and without drug treatment (*Figure 4F*). We observed a lower level of Rac1 activation the basal membrane compared to the apical membrane after application of PMA or scoloplamine, suggesting a potential negative regulation of Rac1 at the basal membrane. This may be consistent with the critical role of Rac1 in intestinal crypt patterning and differentiation (*de Santa Barbara et al., 2003*).

## Benchmarking galene using in vivo and ex vivo intestinal crypt imaging data

To benchmark the core motion correction algorithm used in *Galene* using real data, we exported the intensity of each frame from the intestinal crypt FLIM data shown in *Figure 4* as time series data. Each frame has relatively low signal to noise and we found that the SIFT algorithm was unable to reliably extract feature points from the data to use for realignment. We therefore assessed the performance of StackReg, SIMA and *Galene*. *Figure 4—figure supplement 2* shows the results of the realignment of intensity only versions of the intestinal crypt data acquired (A) in vivo through an optical window and (B) ex vivo. Due to the rapid motion observed in this image, the linear transformation used by StackReg is not able to adequately correct for the motion and the correlation between each frame and the reference frame is not improved compared to the unaligned data. SIMA provides an improvement in the image quality compared to the unaligned data and an increase in the average correlation between frames; however, a substantial motion artefact is still visible in the integrated image. *Galene* produces a significant improvement in the image quality over SIMA and StackReg. In line with the results of our simulations and in addition to enabling correction of time resolved data, *Galene* demonstrates a significant improvement in the realignment of both in vivo and ex vivo imaging data.

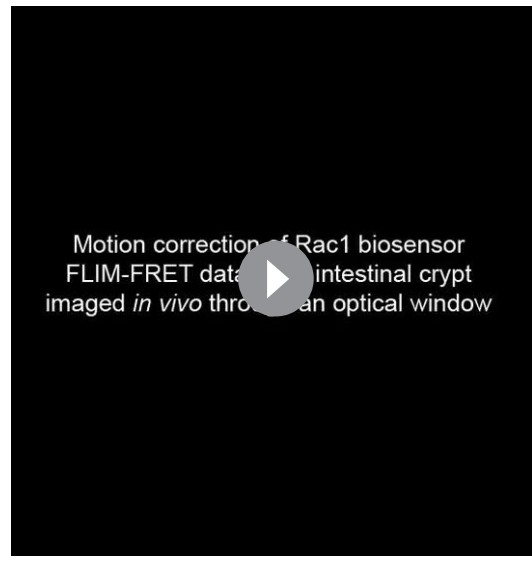

**Video 2.** Motion correction of Rac1 biosensor FRET of an intestinal crypt imaged in vivo through an optical window. Associated with *Figure 3*.

DOI: https://doi.org/10.7554/eLife.35800.014

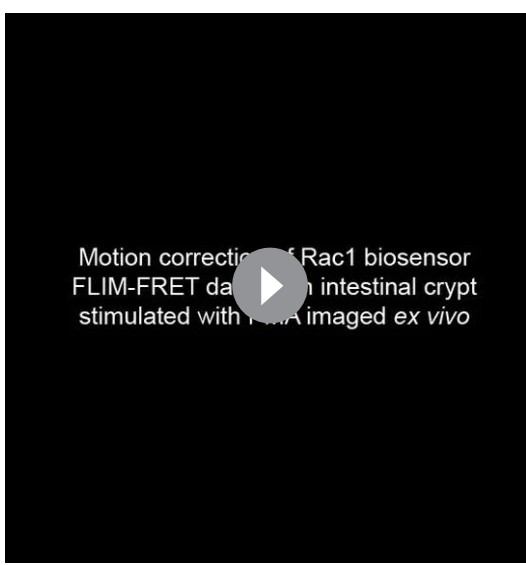

**Video 3.** Motion correction of Rac1 biosensor FRET in an intestinal crypt stimulated with PMA imaged ex vivo. Associated with *Figure 3*.
DOI: https://doi.org/10.7554/eLife.35800.015

## Functional and structural longitudinal imaging during early adhesion events in an intrasplenic model of pancreatic cancer metastasis through an optical window

In a recent study, we demonstrated that transient 'priming' using the pharmaceutical Rho kinase inhibitor Fasudil in a model of pancreatic cancer (PC) improved response to chemotherapy and impaired metastasis to the liver in an intrasplenic model (*Vennin et al., 2017*). Src kinase has been shown to play a critical role in cell adhesion and proliferation in cancer (*Brunton and Frame, 2008*) and is potentially an anti-invasive target in PC (*Evans et al., 2012*; *Morton et al., 2010b*; *Nobis et al., 2013*). We previously observed a reduction in Src kinase activity in in vitro models of invasion and in endpoint xenograft models of a primary tumour imaged using a skin flap technique. We therefore hypothesised that the reduction in the number of metastases after priming with Fasudil could be, in part, a consequence of a reduction in early adhesion events caused by disruption of Src activation. However, without correction for sample motion, the assessment of such early transient events using FRET (illustrated schematically in *Figure 5B*) is not possible in vivo, limiting our ability to quantify colonisation efficiency at this important stage.

We used *Galene* to track Src activity in early adhesion events and so directly assess the effects of Fasudil during early attachment. We injected KPC cells expressing the Src-FRET biosensor (*Wang et al., 2005*) into mice implanted with abdominal imaging windows implanted on top of the liver (see *Figure 5A*) and imaged cells arriving in the liver 4, 8, 16 and 24 hr (see timeline in *Figure 5C*) after injection. We used a multispectral FLIM system to allow us to record the lifetime of the Src biosensor alongside microenvironmental context using a variant of the hyperspectral unmixing approach recently demonstrated (*Cutrale et al., 2017*). *Figure 5Di* shows an example motion corrected merged intensity image in the three spectral channels used and *Figure 5Dii* and iii show the temporal phasor of the 525/50 nm channel and the spectral phasor respectively. Phasor gates associated with Src-FRET, hepatocytes, the vasculature and collagen were identified as shown and used to identify the associated region in the image (*Figure 5Div*). Using the data from these regions, we created a pattern associated with each component and performed non-negative least squares to unmix the autofluorescence signal from nearby liver cells, blood vessels and the collagen network which have distinct hyperspectral signatures. To determine the lifetime of the Src biosensor, we identified regions containing the biosensor using phasor analysis and fitted the data to a single exponential model in the donor channel. We manually segmented single cells to determine the average lifetime per cell.

The liver shows significant motion when imaged behind an optical window due to its proximity to the lungs and heart and attachment to the diaphragm. This frustrates attempts to acquire data for lifetime and hyperspectral unmixing as integration over even a few frames leads to significant image blurring as illustrated in *Figure 5Ei*. *Figure 5Eii* shows the same image after correction for motion with liver cells shown in grey, blood vessels in red, collagen in magenta. The lifetime of the biosensor is color-coded from blue (low lifetime, low Src activity) to red (high lifetime, high Src activity, see schematic). Using *Galene*, we were able to reliably correct for motion in this context and so probe the activation of Src in relation to the true attachment state or spreading phenotype of cells during these early adhesion events in the liver. Here, we saw a significant increase in Src activity (increase in Src biosensor lifetime) after 8 hr, which was maintained up to 16 hr before plateauing after 24 hr

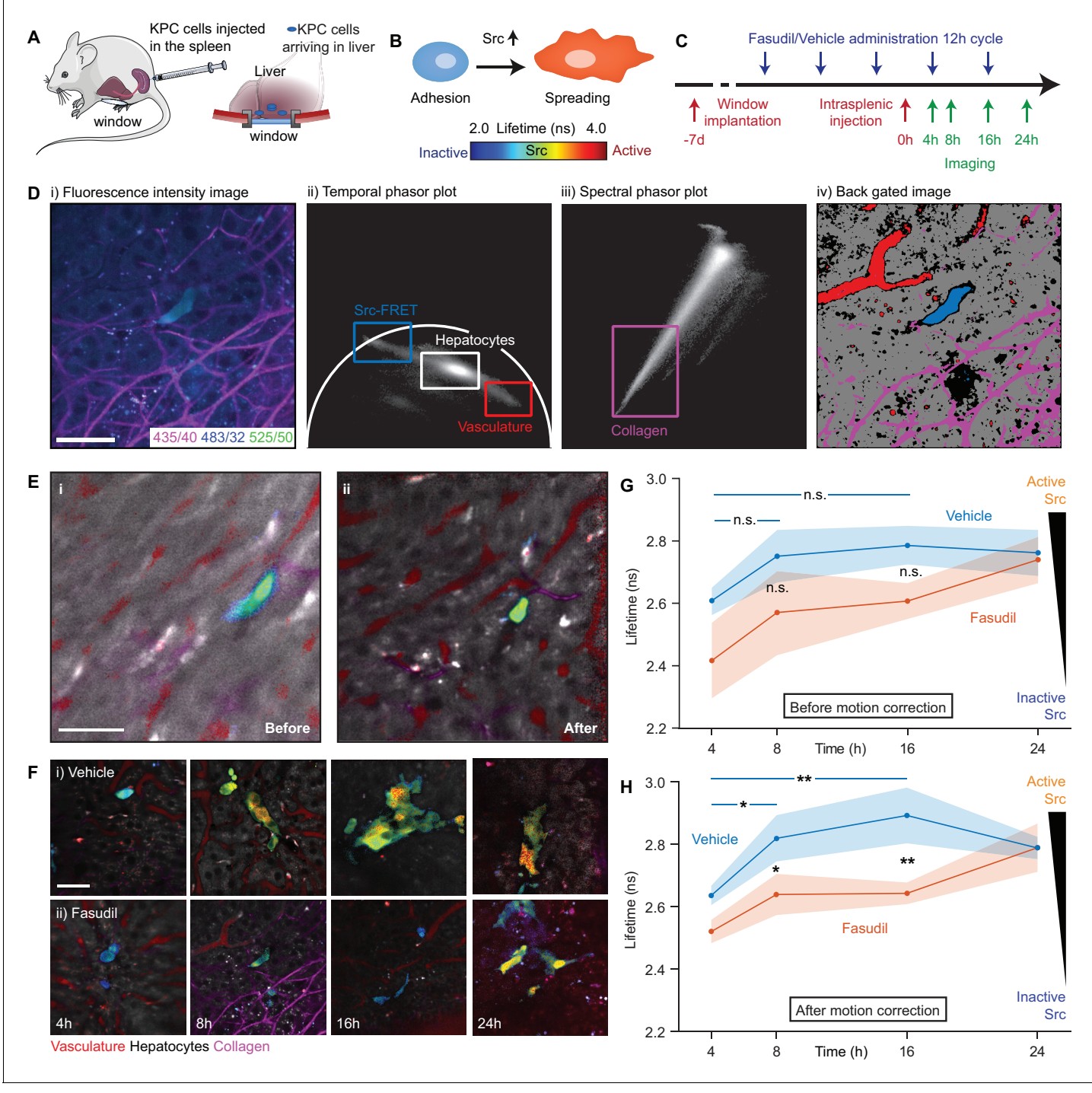

**Figure 5.** Longitudinal imaging of Src activity in cancer cells colonising the liver in an intrasplenic model of pancreatic cancer metastasis with micro-environmental context in response to priming with Fasudil. (A) Cartoon of intrasplenic model of pancreatic cancer using KPC cancer cells expressing a Src-FRET biosensor. (B) Cartoon of morphology of KPC cells during early attachment events. (C) Experimental timeline showing timings of window implantation, intrasplenic injection, Fasudil (or vehicle) administration and imaging. (D) Illustration of identification of fluorescence components; (i) merged spectral intensity image, (ii) temporal phasor plot of 525/50 nm channel with gates used to select fluorescence components, (iii) spectral phasor plot, (iv) back projection image showing gates highlighted in (ii) and (iii). (E) Merged Src-FRET biosensor lifetime and hyperspectral unmixing image **i** before and (ii) after motion correction acquired 8 hr after intrasplenic injection. Cancer cells expressing the Src biosensor are colour-coded using the rainbow lifetime scale. Autofluorescence contributions obtained using hyperspectral unmixing shown in (red) vasculature, (grey) hepatocytes and (magenta) collagen. (iii) Estimated displacement traces in (blue) x and (red) y directions over time. (iv) Correlation between reference frame and (red) uncorrected and (blue) corrected images over time. (F) (i,ii) Example images showing Src activity and micro-environmental context in mice treated with

*Figure 5 continued on next page*

*Figure 5 continued*

(i) vehicle and (ii) 100 mg/kg Fasudil according to the timeline show in (C) at 4, 8, 16 and 24 hr after intrasplenic injection respectively. (G,H) Average Src biosensor lifetime in cancer cells colonising the liver in response to Fasudil treatment, using images (G) before and (H) after motion correction. n = 3 mice per condition, 20–45 cells per time point. Results show means ± SEM (shaded). *p* values were determined per-mouse by unpaired t-test, *p<0.05; **p<0.01. Mouse and liver illustrations were adapted from Servier Medial Art, licensed under the Creative Commons Attribution 3.0 Unported license.
DOI: https://doi.org/10.7554/eLife.35800.016

The following source data is available for figure 5:

**Source data 1.** Source data for graphs show in *Figure 5H and G*, showing average Src-FRET biosensor lifetimes at 4, 8, 16 and 24 hr after intrasplenic injection per mouse, (Sheet 1) before motion correction and (Sheet 2) after motion correction.
DOI: https://doi.org/10.7554/eLife.35800.017

upon spreading (see *Figure 5E*, control situation). In line with our previous study (*Vennin et al., 2017*), to mimic systematic ROCK inhibition or adjuvant therapy in the presence of circulating tumour cells, we treated mice with Fasudil at 12 hr intervals with three treatments before intrasplenic injection (see timeline in *Figure 5C*). Mice treated with Fasudil exhibited a significantly reduced and delayed Src activity, in line with a delayed spreading phenotype, compared to those treated with the vehicle (see *Figure 5Fi*, blue-green shift at 8 hr and *Figure 5Fii*, blue-green shift at 24 hr, quantified in *Figure 5G*), indicating that Src-dependent spreading and activation during the first attachment events in the liver are indeed impaired by treatment with Fasudil. These results demonstrate a new role of Fasudil priming in altering adhesion efficiency in secondary sites (*Rath et al., 2017*; *Vennin et al., 2017*). There is an urgent need to develop anti-metastatic treatments in PC and other metastatic cancer types (*Steeg, 2016*) and functional intravital microscopy combined with *Galene* may help development of new strategies to monitor agents that affect this critical event preceding colonisation (*Ritsma et al., 2012*; *Steeg et al., 2011*).

We note that aside from reducing the image quality, motion during acquisition can have a number of more subtle effects: (1) blurring of the biosensor fluorescence with background autofluorescence which may have a very different lifetime can artificially change the apparent lifetime of the biosensor, giving a misleading result and (2) motion can significantly distort the apparent shape of the cells. Both artefacts are observed here; we see a reduction in the apparent lifetime of the biosensor due to blurring with the low autofluorescence lifetime of surrounding liver cells and an artefactual elongation and apparent spreading of the cancer cell (compare *Figure 5Di–ii*) which may lead to an incorrect assumption about their attachment state (see *Figure 5B*). *Video 4* shows the accumulated frames before and after correction (note that *Video 4* shows the mean arrival time of all channels acquired, not just the Src biosensor donor). To evaluate the impact of motion correction on our ability to quantify Src activity in dataset, we analysed the lifetime of the uncorrected data in the same way (*Figure 5H*). The blurring of the biosensor lifetime with the background

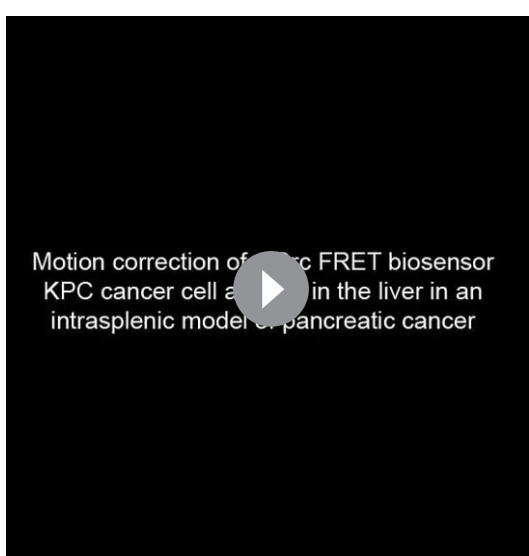

**Video 4.** Motion correction of KPC Src-FRET biosensor cells and liver autofluorescence imaged in an intrasplenic experiment through optical window. Associated with *Figure 5D*. (*top left*) Each intensity frame from the FLIM acquisition (*green*) superimposed on the reference frame (*magenta*). (*top right*) The motion compensated frames (*green*) superimposed on the reference frame (*magenta*); frames which have been excluded due to a poor correlation are marked as such. (*bottom left*) Cumulative intensity merged FLIM lifetime image without motion correction, estimated using the first moment of the decay. (*bottom right*) Cumulative intensity merged FLIM lifetime image with motion correction, estimated using the first moment of the decay.
DOI: https://doi.org/10.7554/eLife.35800.018

autofluorescence leads to an overall reduction in the lifetime in all treatment conditions. As the magnitude of this effect varies greatly from cell to cell depending on the motion and the environment of each cell, we found a significantly higher degree of variance within each condition. This increased variability abolishes our ability to statistically distinguish the conditions, highlighting the importance of motion correction to obtain robust results in this context (compare in *Figure 5G–H*).

## Motion correction of clinical autofluorescence imaging of human skin in three dimensions

The lifetime of NADH and FAD autofluorescence can be used as a readout of metabolic activity (*Blacker and Duchen, 2016*; *Lakowicz et al., 1992*; *Skala et al., 2007*) with potential applications in the detection of precancerous tissue. This autofluorescence signal has been investigated in a clinical context for diagnostic purposes (*König, 2012*) using static (*Patalay et al., 2012*), flexible (*König et al., 2008*) and handheld multiphoton (*Sherlock et al., 2015*) microscopes. Unlike optical window experiments, where the inverted imaging configuration and weight of the mouse largely constrains the sample motion to two dimensions, the movement observed imaging human skin in an upright configuration occurs isotropically in three dimensions and so correction for lateral motion alone is insufficient. We therefore demonstrate two approaches for handling motion in three dimensions: (1) real-time detection and compensation for axial sample motion when imaging a single plane and (2) 3D motion correction within a z-stack. We recently demonstrated (*Sherlock et al., 2015*) a handheld multiphoton microscope system that incorporates an active (online) axial motion compensation system that corrects for motion in z-axis in real time. The optical coherence tomography (OCT)-based correction system tracks sample motion perpendicular to the imaging plane in real time and adjusts the objective position to keep the selected plane in focus. We applied this system to collect short FLIM images of human epidermis; *Figure 6Ai–iii* shows the autofluorescence FLIM images (i) without motion compensation, (ii) with axial motion compensation alone and (iii) both axial and lateral motion compensation with *Galene*. The combination of active axial motion compensation and software-based lateral motion compensation is able to effectively remove the motion artefact observed.

In a clinical setting, it is often desirable to obtain a 3D map of the autofluorescence lifetime to build up structural and functional information resolved into the strata of the skin, for example to quantify drug penetrance and delivery (*Roberts et al., 2011*). We acquired 3D FLIM images of autofluorescence from the dorsal forearm of a volunteer using a commercial clinical FLIM instrument (*Leite-Silva et al., 2016*). Capturing time-resolved depth stacks with sufficient photon counts can be a time-consuming process; a 30–50 μm stack may take 10–15 min to acquire. A certain degree of motion during this period is inevitable in live subjects and, since conventionally each frame is accumulated consecutively (*Figure 6Bi*), this can lead to both blurring of individual images in the stack and displacement between images in the stack. To overcome this issue, we accumulated a number of scans over the entire stack (*Figure 6Bii*). We then apply the same motion estimation approach as used in 2D data, extending the displacement points to 3D. Each stack is aligned to a reference stack. Motion in three dimensions during each stack acquisition can then be estimated and corrected. To enable correction of these large volumes, we used GPU computation and several algorithmic optimisations to reduce the processing time (see Methods, *Figure 7—figure supplement 1*). We acquired 50 μm stacks with 36 images, accumulating 10 stacks in total. *Figure 6C* shows the autofluorescence FLIM images before and after image-based 3D motion compensation. We see that we are able to track and correct for motion in three dimensions during the stack acquisition, obtaining undistorted deep-tissue data free from motion artefacts. These approaches may enable the use of autofluorescence imaging of other parts of the body more susceptible to sample motion in three dimensions such as the chest.

## Motion correction of three dimensional multispectral intensity data

*Galene* can also be used to correct for motion in time lapse fluorescence microscopy data, supporting data import and export from a number of common microscopy formats, OME-TIFF (*Goldberg et al., 2005*) and Imaris data formats. We applied *Galene* to intravital multi-channel 3D imaging of immune cells in the inguinal lymph node (*Suan et al., 2015*) and benchmarked its performance against drift correction in Imaris. Intravital imaging is a crucial tool in immunology, providing

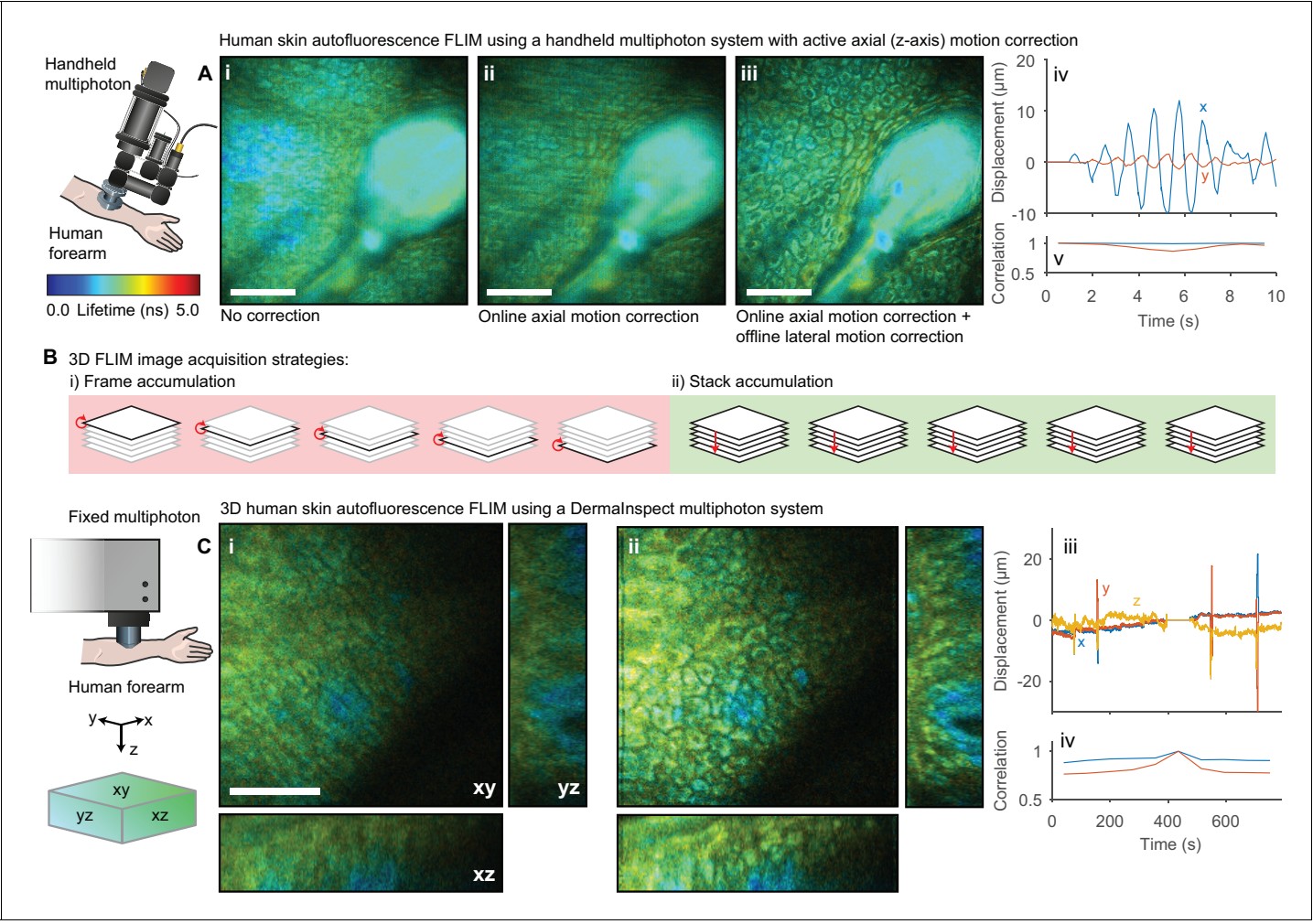

**Figure 6.** Motion correction of autofluorescence imaging of human skin in vivo. (**A**) Example intensity merged FLIM image of human skin around a hair follicle on a handheld multiphoton microscope with an active axial motion correction system, (i) without motion compensation, (ii) with axial motion correction and (iii) with axial and lateral motion compensation. White scale bars 50 μm. (iv) Estimated displacement traces in (*blue*) x and (*red*) y directions over time. Correlation between reference frame and (*red*) uncorrected and (*blue*) corrected images over time. (**B**) Cartoon illustration of (i) conventional frame-accumulation 3D stack acquisition strategy where frames from each slice in the volume are accumulated in turn and (ii) stack based accumulation where multiple acquisition of the entire stack recorded in a single pass are acquired, enabling motion correction in three dimensions. (**C**) Example intensity merged orthogonal projection of a 3D volume of human skin acquired on a commercial DermaInspect multiphoton microscope (i) before and (ii) after motion correction showing xy, yz, and xz sections through the volume. White scale bars 100 μm. (iv) Estimated displacement traces in (*blue*) x and (*red*) y directions over time. (v) Correlation between reference frame and (*red*) uncorrected and (*blue*) corrected images over time. Data shown in (**A**) were acquired at Imperial College, London (*Sherlock et al., 2018*) and reanalysed with kind permission under the Creative Commons Attribution 4.0 International licence. Mouse and liver illustrations were adapted from Servier Medial Art, licensed under the Creative Commons Attribution 3.0 Unported license.

DOI: https://doi.org/10.7554/eLife.35800.019

unique spatiotemporal information about the localisation, function and interactions between immune cells in their native environment. The organisation and migration of different classes of immune cells within the lymph nodes has been shown to play a critical role in the adaptive immune response (*Kastenmüller et al., 2012*). For example, the production of antibodies in response to antigen re-exposure after vaccination depends on the interaction between CD4[+] T cells and B cells at a number of specific locations in the lymph node. Tracking of the migration of these cells is therefore critical to understanding this process and its dysregulation. We imaged tdTomato labelled B cells (red), Kaede CD4[+] OT2 T cells (green) and subcapsular sinus macrophages labelled with Alexa680 (magenta) in a 150 μm z-stack through the inguinal lymph node (SHG signal from fibrillar capsule, blue) over 30 min

(*Figure 7A*). Over the time series, the macrophages and capsule are essentially static while there is significant migration of the CD4$^+$ T cells. Over this period, substantial sample motion is observed; there is a slow drift due to slight shifts in the immersion liquid meniscus and faster displacements due to physiological motion, primarily respiration. These displacements are visible in the temporally

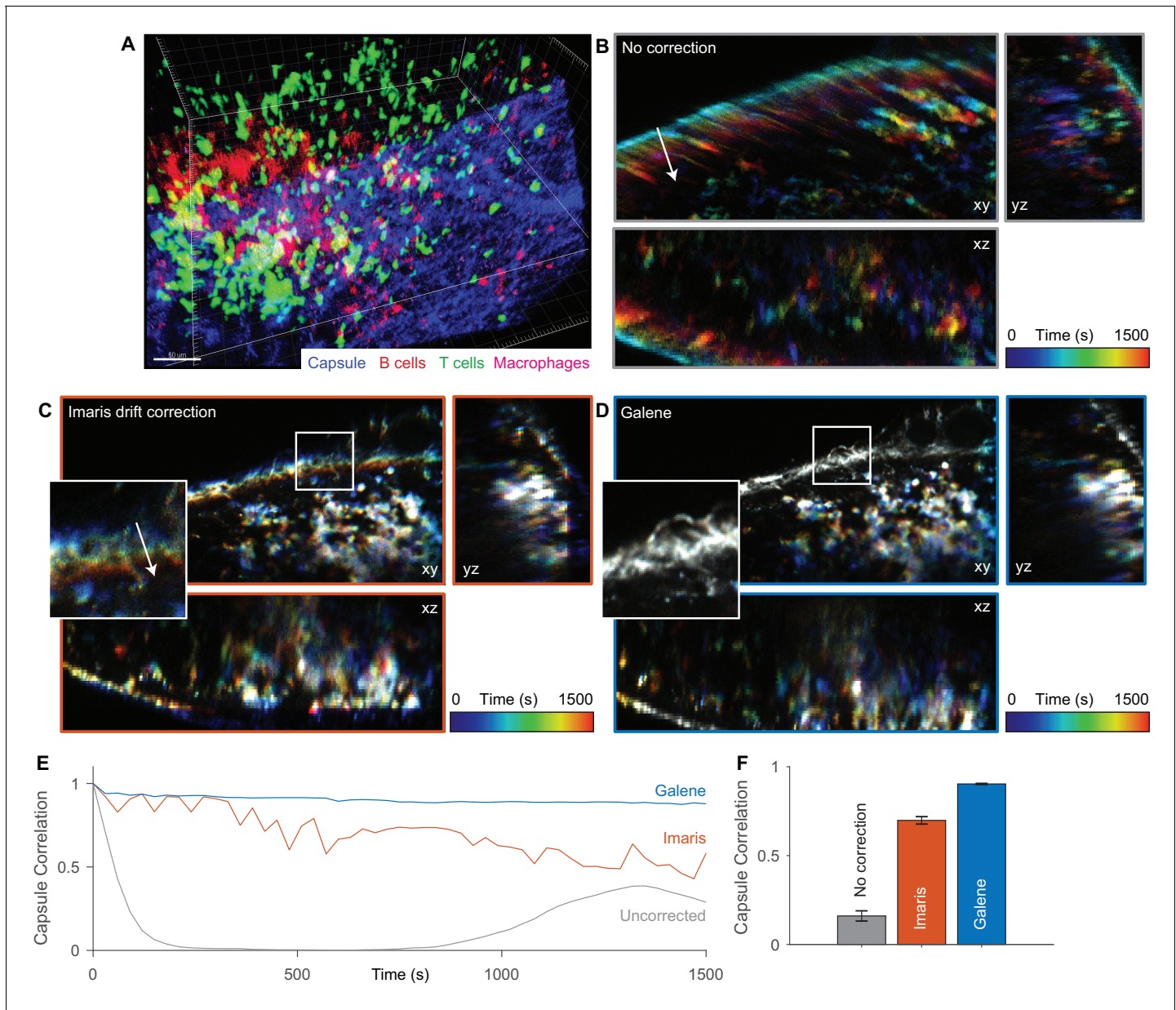

**Figure 7.** Motion correction of multispectral 3D imaging of labelled immune cells in a murine lymph node. tdTomato labelled B cells (*red*), Kaede labelled OT2 T cells (*green*) and subcapsular sinus macrophages labelled with Alexa 680 (*magenta*) imaged in a 150 μm z-stack through the inguinal lymph node (SHG signal from fibrillar capsule, *blue*) over 30 min. (A) One time point rendered volumetrically. (B–D) Temporally colour-coded (*blue*, early time points; *red*, late time points) orthogonal projections of time series (with spectral channels merged) with (B) no correction, (C) Imaris drift correction and (D) motion correction with Galene. Inset, expanded view of region highlighted in white. (E) Correlation between the stationary collagen and macrophage signal for each volume in the sequence and the reference volume for each package. (F) Average correlation for each correction package. Results show mean ± SEM, calculated per time point.

DOI: https://doi.org/10.7554/eLife.35800.020

The following figure supplement is available for figure 7:

**Figure supplement 1.** Algorithmic optimisation for realignment of three dimensional data.

DOI: https://doi.org/10.7554/eLife.35800.021

colour-coded projections shown in *Figure 7B*, where early time points are shown in blue and late time points in red. We used the Imaris 'spot tracking' function to track the stationary macrophages and used their trajectory to correct for drift. This provides a reduction in the motion artefact, however the correction is not complete (see *Figure 7C* inset, *Video 5*), and there is a gradual loss in the correlation between the nominally static capsule signal over time (*Figure 7E and F*). We then used *Galene* to correct for the motion based on the static capsule and macrophages and observed a significant improvement in the quality of the correction (see *Figure 7D*, *Video 5*). This will allow more accurate quantification of in vivo cell movement within this organ over long imaging periods even in the presence of physiological motion and sample drift.

## Discussion

We developed *Galene*, an open source tool to correct for sample motion in two and three-dimensional intravital functional imaging data where many frames are integrated to provide sufficient signal to noise to accurately determine, for example, the activity of FRET reporters using FLIM. The source code for *Galene* is freely available alongside compiled executable for Windows and Mac at http://galene.flimfit.org/.

   *Galene* uses a fitting approach that explicitly accounting for the raster scan pattern performed by laser scanning microscopes to determine sample motion both between frames and during each frame. We use the Lucas-Kanade framework to efficiently estimate the sample displacements following the approach of *Greenberg and Kerr (2009)*. Using the motion estimate, both FLIM and intensity-based data can be reconstructed, allowing quantitative analysis free from artefacts introduced by motion. To scale this approach to motion correction across 3D volumes where there may be thousands of displacement points, we implement two modifications. First, we take advantage of graphical co-processors to accelerate computationally intensive sections of the motion correction algorithm using to take advantage of graphical co-processors. We then exploit the structure of the optimisation problem to radically reduce the computational burden of estimating each optimisation step. These modifications reduce the time required to perform motion correction by a factor of 30 for typical volumetric datasets and open the door to online motion correction in the future.

   Using simulations of FLIM data, we explored the range of motions that can be effectively compensated with *Galene*. As expected, we found that motion perpendicular to the scanner fast axis – a user controllable parameter on most modern confocal microscopes – can be more effectively compensated than those along the slow axis. We found that motions covering a broad range of physiologically relevant motions, from respiration to the heartbeat can be effectively compensated when the magnitude of the motion was ~10% of the FOV, while significantly large motions, up to 30% of the FOV could be corrected at lower frequencies. We note that faster motions still may be compensated by using faster scan rate, for example by employing a resonant scanner. The difference in quality of correction of motions in the fast and slow axis is due to the frequency with which different positions are sampled during the acquisition. Acquiring a 256 × 256 image at 1000 Hz means that every x-position will be sampled every millisecond (albeit at a different y position), while every y-position is sampled every 256 milliseconds. A motion purely in the fast scan (x-) direction will appear as a wave-like motion while a motion in the slow axis will appear to compress and stretch the image and, for sufficiently large motion, whole lines may be missed. This means that the data constrain the estimate of motion in the fast axis more strongly than in the slow axis.

   We benchmarked the core motion correction algorithm against existing open source packages by generating simulated fluorescence intensity time series data and found that *Galene* can correct for a significantly larger range of motions than these packages. We confirmed these results using intravital FLIM-FRET biosensor imaging data and multispectral intensity-based data to demonstrate that we can compensate for a range of physiological motions. We showed that we can correct for motion occurring both

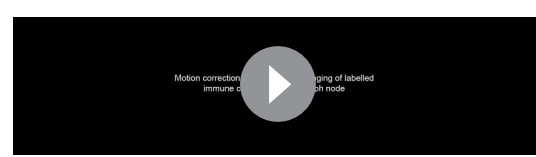

**Video 5.** Motion correction of multispectral imaging of labelled immune cells in a murine lymph node. Associated with *Figure 7*. (*left*) Uncorrected time series. (*middle*) Time series corrected using Imaris drift correction (*right*) Time series corrected with Galene.
DOI: https://doi.org/10.7554/eLife.35800.022

between frames and within frames which otherwise render the image unintelligible and thereby retrieve both cellular and subcellular resolution. This will enable researchers to apply functional imaging modalities in contexts previously inaccessible due to excessive motion and therefore extend acquisition times, providing higher signal to noise (or using a lower excitation power to prevent tissue damage or photo-bleaching). Other recent applications of intravital FRET investigating, for example, RhoA (*Nobis et al., 2017*), intricate Erk signalling propagation events from in the skin (*Hiratsuka et al., 2015*) or cancer stemness (*Kumagai et al., 2015*), PKA in vascular permeability (*Yamauchi et al., 2016*) and stromal targeting in melanoma (*Hirata et al., 2015*) could benefit from this approach. We contrasted our approach with pharmacological inhibition of motion using scopolamine in sections of ex vivo intestine. We observed that, although scopolamine effectively inhibited sample motion, it induces a significant activation of Rac1. This would compromise any study of Rac1 signalling in this context, illustrating the potential pitfalls of pharmacological approaches to inhibiting sample motion.

We used *Galene* to image the activation of Src, a key regulator of metastasis in PC (*Erami et al., 2016*; *Evans et al., 2012*; *Nobis et al., 2013*; *Vennin et al., 2017*), via FLIM-FRET in cancer cells arriving and attaching in the liver in an intrasplenic metastasis model of pancreatic cancer using live longitudinal imaging with titanium windows. We previously demonstrated that ROCK inhibition with Fasudil improved response to standard-of-care chemotherapy and reduced metastasis of pancreatic cancer cells using intrasplenic and orthotopic models and have shown that Src is a key regulator of metastasis in PC (*Erami et al., 2016*; *Evans et al., 2012*; *Nobis et al., 2013*; *Vennin et al., 2017*). To date, our ability to use FLIM-FRET imaging to observe these early, first attachment events has been hindered by sample motion. The extensive physiological motion observed in the liver not only significantly degrades the image but also, by mixing the FRET signal with the autofluorescence background, introduces a significant artefact into the FLIM readout of the Src-FRET activity. Using *Galene*, we observe a significant delay in Src activity after priming with Fasudil at early time points, providing a potential mechanism for the significant reduction in metastatic colonisation observed in endpoint metastatic burden experiments (*Vennin et al., 2017*). As highlighted above, this may have wider applications in other metastatic cancers where early adhesion events are difficult to study.

Imaging human skin autofluorescence using two clinical multiphoton systems we found that, unlike murine optical window experiments, where the weight of the mouse tends to constrain sample motion laterally within the image plane, in a clinical context sample motion occurs isotropically in the x, y and z-axis. Motion in the z-axis cannot be corrected when imaging at a fixed focal plane as the sample physically moves out of the scan area. We demonstrated two methods for motion correction in three dimensions; we first combined *Galene* with a custom handheld multiphoton system incorporating axial motion correction (*Sherlock et al., 2015*) while imaging human skin. By combining online axial motion compensation with lateral motion using *Galene* we can effectively compensate for motion in three dimensions. In an alternative approach, we acquired z-stacks of human skin, frequently used, for example, in skin penetration studies (*Labouta et al., 2011*). Here, integration times on the order of tens of minutes are not uncommon due to weak autofluorescence signal and excitation powers limited by safety considerations. We showed that, by generalising the 2D motion correction approach to estimate motion in a three dimensional raster scan, we can effectively correct for isotropic motion in 3D. This will enable longer and more accurate FLIM volume acquisitions and imaging in locations more susceptible to physiological motion such as the chest where respiration is a significant challenge.

We have demonstrated that it is possible to robustly analyse motion corrected FLIM data using several methods which require a high signal noise levels and sample stability using data from two widely available commercial FLIM systems. *Galene* may be applied directly to data acquired on systems post capture and may even be applied retrospectively to existing data. This approach could be readily applied to confocal endoscopic systems (*Kennedy et al., 2010*; *Siegel et al., 2003*; Sparks et al., in press; *Sun et al., 2013*) to allow TCSPC FLIM acquisitions in an intraoperative context, for example for tumour margin assessment (*Gorpas et al., 2015*; *Wang et al., 2017*). For such applications, this approach could readily be extended to enable real time correction of FLIM data, as the processing time is typically significantly shorter than data acquisition, even for high speed 3D applications. In principle, this system could be combined with a hardware-based lateral motion correction approach (*Sherlock et al., 2018*), that can track large displacements to enable high fidelity

correction when the microscope itself is moving. This would allow, for example, a macro scale FLIM map to be constructed by freely moving the imaging system across the sample.

In addition to correction of time resolved fluorescence data, we also demonstrated that Galene can effectively correct for motion in fluorescence intensity time series data in both two and three dimensions using multispectral intravital imaging data.

*Galene* will therefore allow researchers to apply time resolved functional imaging in a broader range of contexts, relaxing previous restrictions on sample stability and imaging duration and make use of data which would have previously been discarded, in vitro, in vivo and in the clinic.

# Materials and methods

## Key resources table

| Reagent type (species) or resource | Designation | Source or reference | Identifiers | Additional information |
|---|---|---|---|---|
| Strain, strain background (Rac1-FRET, C57BL/6 (mixed)) | Rac1-FRET biosensor mouse | *Johnsson et al. (2014)* | Rac1-FRET biosensor mouse | |
| Strain, strain background (BALB/c-Fox1nuAusb, Mus) | BALB/c-Fox1nuAusb | Australian BioResources | BALB/c-Fox1nuAusb | |
| Cell line (Mus) | KPC | *Morton et al. (2010a)* | KPC primary cancer cells | |
| Antibody | active Rac1-GTP | NewEast Biosciences | RRID:AB_1961793 | 1:400 concentration |
| Recombinant DNA reagent | Rac1-FRET biosensor | *Itoh et al. (2002)* | Raichu-1011X ECFP-SEYFP | |
| Recombinant DNA reagent | Src-FRET biosensor | *Wang et al. (2005)*, *Vennin et al. (2017)* | Src-FRET biosensor | |
| Chemical compound, drug | (-)-Scopolamine-N-butylbromide | Sigma-Aldrich | PubChem:CID_6852391; Sigma:S7882 | |
| Chemical compound, drug | Phorbol myristate acetate | Sigma-Aldrich | PubChem:CID_27924; Sigma:P8139 | |
| Chemical compound, drug | Fasudil | Jomar Life Research | PubChem:CID_163751; Jomar:HA-1077 | |
| Software, algorithm | Matlab R2018a | Mathworks | RRID:SCR_001622 | |
| Software, algorithm | *FLIMfit* | *Warren et al. (2013)* | RRID:SCR_016298 | |

### Motion compensation in TTTR data

### Reconstruction of frames from TTTR data

FLIM data saved in a TTTR format were read in using a custom reader implemented in C++. The TTTR data consist of a stream of events; markers corresponding to the start of frames and beginning and end of lines and photon arrival times. Each event is tagged with a 'macro' time marker, a coarse marker of the time since the experiment start, and photons contain 'micro' time information about their arrival time with respect to the last excitation pulse.

From these markers, the duration of each line, $t_{line}$, the time between the start of each line, $t_{inter-line}$ and the time between the frames $t_{frame}$ are determined. For many microscopes, only around a third of the total line scan time is 'active.' The photons are divided into pixels of equal duration $t_{pixel} = t_{line}/n_{pixel}$. For realignment, the intensity of each frame in the FLIM image is reconstructed, disregarding micro-time information. Using $t_{line}$, $t_{interline}$ and the image dimensions, the macro time of each pixel can then be determined. To reduce noise, an elliptical Gaussian filter with a user-controllable radius in the fast scan axis and a 1px radius in the slow scan axis is applied. This filter is applied to the reference frame and each frame used for realignment but not to the final, realigned data.

### Estimation of sample motion

For each reconstructed frame, we aim to determine the motion during that frame with respect to a user selectable reference frame. We model the sample displacement $D(t)$ using a series of $n$ vectorial

displacements $\{r^{(i)}\}$ equally spaced over the frame time $t_{\text{frame}}$ and assume that the sample moves linearly between these displacements. The sample displacement at time $t$ is then

$$D(t) = r^{(i)} + \left(r^{(i+1)} - r^{(i)}\right)(t - t_i)\frac{n-1}{t_{\text{frame}}}. \tag{1}$$

For a given set of displacements, we can reconstruct the image using subpixel interpolation. For each pixel in the displaced image $\tilde{I}$, the intensity is given by

$$\tilde{I}(x,y,z) = \sum_{i=0,1}\sum_{j=0,1}\sum_{k=0,1} w_x^{(i)} w_y^{(j)} w_z^{(k)} I\left(x + d_x + i,\, y + d_y + j, z + d_z + k\right) \tag{2}$$

where

$$t = x \cdot t_{\text{pixel}} + y \cdot (t_{\text{line}} + t_{\text{interline}}) + z \cdot (t_{\text{frame}})$$

$$d_{x,y,z} = \lfloor D_{x,y,z}(t)\rfloor;\; w_{x,y,z}^0 = 1 - D_{x,y,z} + \lfloor D_{x,y,z}(t)\rfloor;\; w_{x,y,z}^1 = 1 - w_{x,y,z}^0 \tag{3}$$

Note that, unlike the final reconstructed FLIM data, this is not an intensity preserving transformation; instead of displacing each pixel in the input image (potentially leaving gaps in the output image due to motion), we attempt to estimate the intensity for every pixel in the output image. This gives a smoother error function and so improves the convergence.

We wish to determine the optimal set of displacements $\{r^{(i)}\}$ that minimise the sum of square E error between the reference image and the displaced image. We use a variant of the Lucas-Kanade algorithm (*Baker and Matthews, 2004*). The error function is given by

$$E(p) = \sum_{x,y,z}\left(\tilde{I}(x,y,z;p) - T(x,y,z)\right)^2. \tag{4}$$

Where $p = \left\{r_x^{(1)}, r_y^{(1)}, r_z^{(1)} \ldots r_x^{(n)}, r_y^{(n)}, r_z^{(n)}\right\}$ is a vector of the fit parameters. We use a trust region algorithm (*Byrd et al., 2000*) to iteratively minimize the error E. In the trust region method, we use a quadratic approximation to the error function and take steps within a trusted region. The size of the trusted region, $\Delta$ is expanded or reduced depending on how well the actual reduction in the error function agrees with the predicted reduction. To determine the $k+1^{\text{th}}$ step, $\Delta p$, we need to solve the sub problem

$$\min_p\left[E_k + \nabla E_k^T p + \frac{1}{2}p^T H_k p\right] \qquad \text{s.t.} \|p\| < \Delta_k \tag{5}$$

Where $E_k$ is the value of the error function at the $k^{\text{th}}$ step, $\nabla E$ and $H$ and are the Jacobian and Hessian of the error function respectively. We solve the trust region problem using the approach described by Nocedal and Wright (*Nocedal and Wright, 1999*) using the open source dlib (*King, 2009*) implementation.

Using this algorithm, we require analytical estimates of the Jacobian and the Hessian of the error function (*Equation 4*) at each step. Using the formulation above the Hessian depends on the last displacement parameters and so must be recomputed at each step. This imposes a significant computational burden. Instead of displacing the image to match the reference at each step, we can conceptually imagine displacing the reference to match the last shifted image. We then invert the displacements and apply them to the image and iterate. This approach is known as the 'inverse compositional' approach (*Baker and Matthews, 2004*). In this formulation, the error function for the update step $\Delta p$ is

$$E'(p + \Delta p) = \sum_{x,y}\left(\tilde{I}(x,y;p) - \tilde{T}(x,y;\Delta p)\right)^2. \tag{6}$$

This formulation has been demonstrated to be equivalent to the conventional Lucas-Kanade approach (*Baker and Matthews, 2004*) in terms of convergence, but significantly reduces the computational complexity of each update step.

The Jacobian $\nabla E'$ w.r.t the parameter updates is given by

$$\nabla E' = \sum_{x,y,z} 2 \cdot \frac{\partial \tilde{T}(x,y,z;\Delta \boldsymbol{p})}{\partial \Delta \boldsymbol{p}} \cdot \left( \tilde{I}(x,y,z;\boldsymbol{p}) - \tilde{T}(x,y,z;\Delta \boldsymbol{p}) \right). \tag{7}$$

To first order, the derivative of the template image with respect to the parameters is given by

$$\frac{\partial \tilde{T}(x,y,z;\Delta \boldsymbol{p})}{\partial \Delta \boldsymbol{p}} \approx \nabla T(x,y,z) \cdot \frac{\partial \boldsymbol{D}(x,y,z)}{\partial \boldsymbol{p}} \tag{8}$$

Where $\nabla T = \left( \frac{\partial T}{\partial x}, \frac{\partial T}{\partial y}, \frac{\partial T}{\partial z} \right)$ is the gradient of the template image. For the model of motion described above, the partial derivatives of the displacements with respect to $\boldsymbol{p}$ are given by

$$\frac{\partial \boldsymbol{D}_i(x,y,z)}{\partial i_k} = \begin{cases} 1 - \frac{(t-t_k)(n-1)}{t_{\text{frame}}} & tk < t \leq tk+1 \\ \frac{(t-t_k)(n-1)}{t_{\text{frame}}} & tk\text{-}1 < t \leq tk \\ 0 & \text{otherwise} \end{cases}, \quad i \in \{x,y,z\}$$

$$\frac{\partial \boldsymbol{D}_i(x,y,z)}{\partial j_k} = 0, \quad i,j,z \in \{x,y,z\}, \ i \neq j \neq k \tag{9}$$

extending the formulation of Greenberg (*Greenberg and Kerr, 2009*). The Hessian matrix is given by

$$H = \sum_{x,y,z} \left[ \nabla T \frac{\partial \boldsymbol{D}}{\partial \boldsymbol{p}} \right]^T \left[ \nabla T \frac{\partial \boldsymbol{D}}{\partial \boldsymbol{p}} \right] \tag{10}$$

Since the Hessian does not depend on the current values of the parameters, it may be precomputed and reused at each iteration, and indeed between frames. This means that the update step may be performed efficiently, even for relatively large images or numbers of realignment points. Using this formulation, the following parameters may be computed in advance

1. The gradient images $\frac{\partial T}{\partial x}$, $\frac{\partial T}{\partial y}$ and $\frac{\partial T}{\partial z}$ of the template image $T$.
2. The Jacobian of the displacements $\frac{\partial D}{\partial p}$ (*Equation 9*).
3. The steepest decent images $\nabla T \frac{\partial \boldsymbol{D}}{\partial \boldsymbol{p}}$.
4. The Hessian $\boldsymbol{H}$ (*Equation 10*).

The trust region algorithm is performed iteratively until the change in the error function is below a certain threshold, in this case $\Delta E < 10^{-5}$.

## Exploiting structure in the hessian matrix

The computational complexity of computing the error value and the Hessian is $O(n_{px})$. We note that the particular form of the Jacobian (*Equation 6*) means that the Hessian is relatively sparse; increasingly so as the number of parameters increases since the displacement of distant points do not interact.

The computational complexity of each step in the trust region algorithm is dominated by the Cholesky factorization of the Hessian matrix, with is $O(m^3) \approx O\left(n_{px}^3\right)$, since the number of displacement points required scales approximately linearly with the number of pixels for a given motion profile. For 2D problems, the former calculation dominates the computation time, however for even moderate 3D problems, the Cholesky factorization quickly dominates the computation time. For the 3D z-stack timeseries analysed in *Figure 7*, with 51 slices, 51 time points and 15 points per frame (giving 2295 parameter in total), factorisation uses approximately 90% of the computational effort and the realignment process takes 1.5 hours (*Figure 7—figure supplement 1C*).

*Greenberg and Kerr (2009)* organise the parameters $\boldsymbol{p} = \left\{ \left\{ r_x^{(i)} \right\}, \left\{ r_y^{(i)} \right\} \right\}$. In this case, the non-zero entries of the Hessian are organized as shown in *Figure 7—figure supplement 1*. In this case, the non-zero entries are distributed throughout the Hessian. However, if the parameters are

organized according to $\boldsymbol{p} = \left\{ r_x^{(1)}, r_y^{(1)}, r_z^{(1)} \dots r_x^{(n)}, r_y^{(n)}, r_z^{(n)} \right\}$ then the non-zero entries appear as shown in *Figure 7—figure supplement 1B*. In this case, the matrix takes a banded structure with bandwidth $b = 6$. Matrices of this form can be factorized significantly more efficiently (*Martin and Wilkinson, 1965*). We use the lapack routine pbtrf with computational complexity $O(bm^2)$. Using banded factorisation, the realignment of the data in *Figure 7* takes 7.5 minutes (*Figure 7—figure supplement 1B*) and, in this case, the computation time is dominated again by the computation of the error value and Hessian matrix.

## GPU acceleration

Computation of the error value (*Equation 6*) and the Jacobian (*Equation 7*) at a particular set of displacements is a highly parallel task as the contribution from each pixel can be calculated independently. This task is particularly suited for computation on a GPU for a number of reasons: (1) modern GPUs implement 3D interpolation (*Equation 2*) in hardware, a task which can be relatively inefficient to perform on CPUs due to the random and highly strided nature of the memory access pattern. (2) The host-GPU memory transfer during the computation (often the rate limiting step for GPU acceleration) is low as the images can be stored on the GPU at the start of the optimization and do not need to be subsequently updated.

We have implemented a GPU version of the code for calculating both these values. The parameters which may be computed in advance for each optimization are computed on the CPU and transferred to the GPU. *Equations 6 and 7* are implemented on the GPU and a GPU parallel reduction used to compute the respective sums. Since the steepest decent images $\nabla T \frac{\partial \boldsymbol{D}}{\partial \boldsymbol{p}}$ can be very large for a 3D problem (e.g. over 2 Gb for the problem shown in *Figure 7*, too large to fit in many consumer GPUs) these values are streamed to the GPU from the host in parallel with the computation. This allows even large problems to be computed on low to mid-range GPU hardware with 1-2 Gb of memory. Using the GPU routines, realignment of the data in *Figure 7* is reduced by over a factor of 2, to 3.4 minutes.

## Thresholding based on image correlation

To assess the quality of the motion estimation, the correlation coefficient $r$ between each motion corrected frame $\tilde{I}$ and the reference frame $\boldsymbol{R}$ is computed according to

$$r = \frac{\sum_{x,y,z}(\tilde{I}(x,y,z) - |\tilde{I}|)(R(x,y,z) - |R|)}{\sqrt{[\sum_{x,y,z}(\tilde{I}(x,y,z) - |\tilde{I}|)^2][\sum_{x,y}(R(x,y,z) - |R|)^2]}} \tag{11}$$

where $|\cdot|$ denotes the image mean. A threshold $r_{\text{thresh}}$ value may be set such that any motion corrected frames where $r < r_{\text{thresh}}$ are excluded from the reconstructed image. This allows the user to exclude frames where the motion could not be corrected, or the sample has temporarily moved out of the field of view, from the final reconstruction.

## Initial displacement estimates

In any non-linear estimation problem, judicious selection of the initial parameters is important to ensure the global minimum is obtained. Here, we evaluate two potential initial parameter sets. The set yielding the lowest error is used. First, a null displacement $r_{x,y,z}^{(i)} = 0$ is evaluated. Then a three dimensional rigid body displacement estimated using a 3D version of the phase correlation algorithm (*Foroosh et al., 2002*) is evaluated according to Algorithm 1.

## Algorithm 1. Rigid body estimation of displacement

1. Apply a 3D Hanning window to each stack to reduce edge artefacts.
2. Compute the discrete Fourier transform of each image, $F_A$ and $F_B$ respectively, using the fast Fourier transform algorithm.
3. Compute the cross power spectrum $R$,

$$R = \frac{F_A \circ F_B^{'}}{|F_A \circ F_B^{'}|}$$

4. where $'$ denotes complex conjugation and $\circ$ element-wise multiplication.
5. Determine the sub-pixel location of the maximum position of the peak value of the cross-power spectrum by interpolation around the peak.
6. Determine the translation between the two images given by the displacement of the peak location from the origin.

## Reconstruction of motion compensated FLIM data

After computing the displacement estimates, the TTTR data are reconstructed into histogrammed FLIM data taking the estimated sample motion into account. Each photon arrival is assigned to a pixel coordinate $(x, y, z)$ based on the frame, line and (if they exist) pixel markers in the dataset. The photon is then reassigned to the coordinate $\left(x - D_x(t), y - D_y(t), z - D_z(t)\right)$ using the final displacement estimates for the current frame, where $t$ is the macro time relative to the start of the frame. If a correlation threshold has been set, photons arriving during frames where the correlation coefficient is less than the threshold value will be discarded. The sample motion leads to an effective variable integration time across the image. To correct for this, we calculate the integration time by integrating the dwell time in each pixel across the image. This integration time image is saved alongside the data and used to correct the intensity merged FLIM images. This corrects for the variable integration time without altering the photon statistics in the data used for analysis.

## Loading intensity fluorescence imaging data

OME-TIFF data (*Goldberg et al., 2005*) is supported using the OME files C++ implementation (https://github.com/ome/ome-files-cpp), a number of standard microscopy data formats are supported using libbioimage (https://bitbucket.org/dimin/bioimageconvert/). Imaris data are supported using a custom reader implemented in C++ using the HDF5 library. Motion corrected data can be saved to an OME-TIFF or Imaris file respectively.

## Simulation of motion distorted time tagged FLIM data

Monte Carlo simulations of 2D TTTR data distorted by sample motion were performed using a subset of a high SNR, motion-free intensity image $I_s$ of ex vivo pancreas (shown in *Figure 1*). The sample motion was set to a sinusoidal motion such that

$$D_x(t) = A \cos(\theta) \sin(2\pi f t)$$

$$D_y(t) = A \sin(\theta) \sin(2\pi f t)$$

where $A$ and $f$ are the sample amplitude and frequency respectively while $\theta$ is the angle of the motion with respect to the scanner fast axis. A TTTR event stream was simulated according to Algorithm 2.

## Algorithm 2. Simulation of TTTR data

1. Start at pixel $x, y = 0$ at macro time $t = 0$.
2. Determine the sample intensity $\lambda$ at the current pixel, accounting for the sample motion according to $\lambda = \alpha \cdot I_S\left(x + D_x(t), \ y + D_y(t)\right)$, where $\alpha$ is an intensity scaling factor controlling the simulated count rate.
3. Determine the number of photons arriving at this pixel, $N$, drawn from a Poisson distribution with mean $\lambda$.
4. For each photon,
   1. Determine the photon arrival time, drawn from the sum of value drawn from the sum of values drawn from an exponential distribution with mean parameter $\tau$ and a Gaussian distribution representing the instrument response characterized by $\mu = 1.0$ ns and $\sigma = 100$ ps
   2. Determine the macro arrival time by evenly spacing the $N$ photons across the pixel time.
   3. and add to the event stream.
5. Increment the macro time $t$ by the pixel time

6. Move to the next pixel
    1. If moving to the next row
        1. Insert a line end marker,
        2. increment the macro time by the interline time,
        3. insert a line start marker
    2. If moving to the next frame
        1. Insert a line end marker,
        2. increment the macro time by the interface time,
        3. insert a frame start marker,
        4. insert a line end marker
7. Repeat for the specified number of frames.

All simulations shown in *Figure 1* were performed using a frame size of 256 × 256 pixels. The pixel time and interline time were set such that line rate was 1 kHz with a duty cycle of 0.33 and the inter-frame time was set equal to the interline time, approximately matching the scan pattern of the Leica SP8 scanner. The intensity was scaled to produce an average count rate of 1 MHz and 50 frames were generated per image.

## Animal experiments

Animals were kept in conventional animal facilities on a 12 hr day-night cycle and fed *ad libitum*. All experiments were carried out in compliance with the Australian code for the care and use of animals for scientific purposes and in compliance with Garvan Institute of Medical Research/St. Vincent's Hospital Animal Ethics Committee guidelines (ARA 13/17, 16/13, 15/29).

For in vivo Rac1 activity experiments, mice ubiquitously expressing the Raichu-1011X ECFP-SEYFP Rac1 biosensor (*Itoh et al., 2002*), generated previously (*Johnsson et al., 2014*), were used.

## Human experiments

Experiments conducted on healthy human subjects using the DermaInspect were performed with informed consent and approval from the University of Queensland Human Research Ethics Committee (approval number 2007/197–2008001342). Experiments conducted on healthy human subjects using the hand held multiphoton system were performed with informed consent and approval from Imperial College London (approval number 14IC2364).

## Cell culture

Primary KPC cancer cells isolated from $Pdx1$-$Cre$; $LSL$-$KRas^{G12D/+}$; $LSL$-$Trp53^{R172H/+}$ tumours (*Morton et al., 2010a*) were engineered to express a Src-FRET biosensor (*Wang et al., 2005*) modified to replace ECFP with mTurquoise2 using a transposon system (*Vennin et al., 2017*; *Wilson et al., 2007*). KPC cells were cultured in DMEM (Gibco) supplemented with 10% FBS and 1% glutamine, penicillin/streptomycin in 5% $CO_2$.

## Small animal surgery and imaging

### Implantation of and imaging through optical windows

The application of optical imaging windows in in vivo imaging and their implantation into the peritoneal wall were described in detail previously (*Ritsma et al., 2013*). Prior to the surgery and up to a mininum of 72 hr afterwards the mice were kept on 5 mg/kg of the analgesic Carprofen (Rimadyl) in the drinking water. Mice were weaned of the analgesic 24 hr before imaging took place. Mice were administered buprenorphine (0.2 mg/kg) s.c. immediately prior to and 6 hr post-surgery for further pain control. The imaging window consists of a titanium window ring, onto which a glass coverslip with a diameter of 12 mm was glued with cyanoacrylate one day prior to surgery. The incision site was cleared of hair by shaving and depilation and disinfected with 0.5% chlorhexidine in 70% ethanol. An incision was made down the midline of the peritoneum for imaging of the small intestine or to the left of the midline of the mouse for the imaging of the pancreas. After blunt dissection of the skin surrounding the incision, a purse string suture (Mersilk, non-absorbable silk based) was placed through the skin and muscle of the abdominal wall. To reduce peristaltic and respiration-associated movement of the organs to be imaged in the peritoneum, a drop of cyanoacrylate was placed on the inner ring of the abdominal imaging windows and the organ of interest immobilized at the edge. Positioning was done using sterile cotton gauzes. The windows were then inserted into the incisions

with the skin and the muscle layer placed into the lateral groove of the windows. Finally, the suture was tightened and firmly tied off at the ends. The mice were allowed to recover from the surgery for at least 72 hr, actively foraging, grooming and feeding within minutes after being removed from the anaesthesia respirator. Mice were anesthetized with 3% isoflurane, supplemented with 100% oxygen and were imaged under 1–2% isoflurane supplemented with 100% oxygen on a 37°C heated stage.

## Intrasplenic injection of cancer cells

Prior to intrasplenic injection, abdominal optical windows were implanted in BALB/c-Fox1nuAusb mice on top of the liver, which were subsequently allowed to recover for 1 week. For intrasplenic injection experiments, KPC cells expressing the Src biosensor ($3 \times 10^6$ cells/50 µL HBSS) were injected into the spleens of BALB/c-Fox1nuAusb mice (anesthetized with 3% isoflurane, $O_2$ 1 L/min, vacuum was used constantly to remove excess of $O_2$) as previously described (*Soares et al., 2014*). Mice were subjected to three rounds of priming with 100 mg/kg Fasudil (HA-1077, Jomar Life Research) in saline buffer (or vehicle control) every 12 hr and by oral gavage before intrasplenic injection, and two subsequent rounds of priming with Fasudil to mimic systemic ROCK inhibition during metastatic spread of KPC cells (twice daily administration by oral gavage). The mice were imaged at 4, 8, 16 and 24 hr after injection and sacrificed after the final imaging time point. See *Figure 5C* for treatment timeline.

## Immunisation and antigen trafficking

Kaede OT2 T cells were enriched by negative depletion with biotinylated antibodies for anti-B220 clone RA3-6B2, anti-CD11b clone M1/70, anti-CD11c clone HL3, anti-CD8 clone 53–6.7, and tdtomato SW$_{HEL}$ B cells (*Phan et al., 2003*) were enriched by negative depletion with biotinylated antibodies for anti-CD11b, anti-CD11c, anti-CD4 clone GK1.5, anti-CD43 clone S7 (all from BD Bisociences) and MACs anti-biotin magnetic beads (Miltenyi). $2.5 \times 10^5$ B220$^+$ HEL$^+$ SWHEL tdTomato B cells and V$_\alpha$2$^+$ CD4$^+$ Kaede OT2 cells were adoptively transferred into C57BL/6 recipients and immunised the next day with 20 µg HEL-OVA in Sigma Adjuvant System (Sigma) injected subcutaneously in the lower flank and tail base. To label SCS macrophages, we injected CD169 clone Ser-4 (UCSF Hybridoma Core) conjugated to Alexa Fluor 680 (Invitrogen), 12 hr before imaging. Mice were imaged 7 days after immunisation.

## Ex vivo tissue drug treatment

Sections of freshly excised and flushed duodenal tissue were treated with 1 µM (-)-Scopolamine-N-butylbromide (scopolamine, Sigma-Aldrich, S7882) in PBS for 30 mins at 37°C or 200 nM phorbol myristate acetate (PMA, Sigma-Aldrich, P8139) for 15 mins.

## Image acquisition

### Small animal and ex vivo FLIM imaging

Multi-photon FLIM data were acquired using an inverted Leica DMI 6000 SP8 confocal microscope using a 25 × 0.95 NA water immersion objective on an inverted stage. The sample was excited using a Ti:Sapphire femto-second laser (Coherent Chameleon Ultra II) operating at 80 MHz and tuned to a wavelength of 840 nm. A RLD HyD detector was used with a 483/32 nm bandpass emission filter for FRET biosensor imaging. FLIM data were recorded in single channel mode using a Picoquant Pico-Harp 300 in TTTR mode or using a Cronologic TimeTagger4-2G. For Src biosensor imaging, photon counting was performed in three spectral channels, 435/40, 483/32 and 525/50 nm, using a Cronologic TimeTagger4-2G. Detailed acquisition parameters for all imaging experiments are given in *Supplementary file 1*. Imaging was performed using a heated stage maintained at 37°C.

### Human skin imaging, DermaInspect

Depth resolved in vivo multiphoton tomography of human skin was performed with a DermaInspect system (JenLab GmbH, Jena, Germany), illuminated with an ultrashort (85 femtosecond pulse width) pulsed mode-locked 80 MHz Ti:Sapphire laser (MaiTai, Spectra Physics, Mountain View, USA), tuned to excitation at 760 nm. Emission was collected in four spectral channels using cooled PMTs (PMC-100 Becker and Hickl, Berlin, Germany) with the following spectral filters: 350–450 nm (Channel 1); 450–515 nm (Channel 2); 515–620 nm (Channel 3); 620–670 nm (Channel 4). A TCSPC system

(SPC830, Becker and Hickl, Berlin, Germany) was used to perform FLIM measurements in FIFO mode. A 40 × NA 1.30 Plan-Neofluar oil-immersion (Carl Zeiss, Germany) was used with an in vivo adaptor designed to hold a coverslip against the skin. The space between the in vivo adaptor and the objective lens was filled with index-matching oil. The first spectral channel, 350–450 nm, was used for FLIM analysis.

## Human skin imaging, handheld multiphoton

Autofluorescence images of human skin were acquired using a handheld multiphoton system as previously described (*Sherlock et al., 2018*). Briefly, the sample was excited using a Ti:Sapphire femtosecond laser (Spectra Physics Mai Tai HP) operating at 80MHz tuned to a wavelength of 760 nm coupled to the imaging head through 4 m of NCF. The pulse FWHM at the output of the NCF was approximately 150 fs (*Sherlock et al., 2016*). The sample was imaged using a 60 × 1.2 NA water immersion microscope objective lens (Olympus UPLSAPO60XW). Emission light was separated using a dichroic filter with centre wavelength 705 nm (Smock FF705-Di0) and relayed to a hybrid PMT (Becker and Hickl HPM-100–40) using a fibre bundle. FLIM data were recorded using a Becker and Hickl SPC-830 TCSPC card in FIFO (TTTR) mode. Multiphoton images were acquired with a line scan rate of 256 Hz, allowing a 256 × 256 pixel images to be acquired in 1 s. The system incorporates a hardware axial motion compensation system actively moving the objective using a piezo actuator (PI P-725 PIFOC) in response to the sample axial displacement tracked employing an optical computed tomography (OCT) system operating using a super luminescent diode with centre wavelength of 930 nm (Supremum 930-B-I-10-PM) (*Sherlock et al., 2015*). A volunteer's dorsal forearm was imaged with their arm lying on a flat rigid surface and with the scanner handheld such that the objective was approximately vertical (*Sherlock et al., 2018*). Some of the weight of the scanner was taken by the operator and some was taken by the scanner resting gently on the arm. To introduce axial motion, the volunteer continuously opened and closed their fist with a period of approximately 0.9 s during the acquisition which introduced a change in pressure between the skin and the front surface of the scanner due to the change in the size of the muscle beneath.

## Intravital lymph node imaging

Intravital two-photon microscopy was performed as previously described with some minor changes (*Chtanova et al., 2014*). Briefly, mice were induced with 100 mg/kg ketamine, 5 mg/kg xylazine and maintained on 1–2% isoflurane supplemented with 100% oxygen for anesthesia. Mice were kept warm on a custom heated SmartStage (Biotherm) set to 37°C. The inguinal lymph node was mobilised along with the intact inguinal ligament in a skin flap and fixed on a base of thermal conductive T-putty (Thermagon Inc.) with VetBond tissue glue (3M). The cortical surface of the lymph node was exposed by microdisseting the skin and overlying fat and fascia layers. Imaging was performed on a Zeiss 7MP two-photon microscope (Carl Zeiss) powered by a Chameleon Vision II ultrafast Ti:Sapphire laser (Coherent Scientific). Images were acquired with a W Plan-Apochromat 20 × 1.0 NA DIC (UV)Vis-IR water immersion objective. Excitation wavelengths used were 870 nm, to detect KD green and tdtomato red. Fluorescent images were acquired with a LBF 760 and BSMP 760 to enable detection of far-red signals. Non-descanned detectors were SP 485 (blue; SHG), BP 500–550 (green; KD), BP 565–610 (red; tdTomato) and BP 640–710 (far-red; Alexa Fluor 680).

## Data analysis

### Lifetime analysis of FLIM data

Raw and aligned FLIM data were analyzed in *FLIMfit* (*Warren et al., 2013*) using a maximum likelihood estimator. A 3 × 3 smoothing kernel was applied to the data before analysis. Instrument response functions (IRFs) were determined using a reference dye measurement as previously described (*Conway et al., 2017*). The data were fitted to a single exponential model to determine the average fluorescence lifetime.

### Complex donor FRET analysis of Rac1 biosensor intestinal crypt data

The data were fitted to a FRET model accounting for the complex decay profile of the ECFP donor as presented in (*Warren et al., 2013*). The complex decay of ECFP originates in the existence of two major conformations of ECFP with different lifetimes (*Demachy et al., 2005*). In this model, we

assume the ECFP Rac1 biosensor exists in two conformations, associated with high and low FRET activity respectively. In each of these conformations, there is a mix of ECFP conformations, which do not affect the overall biosensor conformation. In this model, the FRET efficiencies for the two ECFP conformations are linked by

$$\frac{E_1}{1-E_1} = \frac{\tau_1}{\tau_2} \cdot \frac{E_2}{1-E_2} \tag{12}$$

and so the total decay for a given biosensor conformation is

$$F(t; E_1) = \sum_{i=1}^{2} \beta_i \exp(\tau_i(1-E_i)) \tag{13}$$

We can then fit our data to a model consisting of two biosensor conformations with a high FRET activity $E_1^H$ and low FRET activity $E_1^L$ respectively

$$I(t; E_1^H, E_1^L) = I \cdot \left(\gamma_H F(t; E_1^H) + (1-\gamma_H) F(t, E_1^L)\right) \tag{14}$$

where $\gamma_H$ is the fraction of biosensor in the active conformation in a given pixel.

To determine the lifetimes of the two ECFP conformations, we made time resolved measurements of KPC cells expressing ECFP alone. By fitting globally to a bi-exponential decay, we found ECFP was best fit by two components $\tau_1 = 1330$ ps and $\tau_2 = 3350$ ps with fractional contribution of the first component $\beta_1 = 0.434$. We fixed these values and fitted the measured Rac1 FRET data to the model described in *Equation 14* globally to obtain the values of the FRET efficiency for the high- and low-activity conformations, $E_1^H$ and $E_1^L$. We used these values to determine the fraction of active biosensor in each pixel.

## Phasor analysis of FLIM data

Phasor analysis of FLIM data was performed following (*Digman et al., 2008*). The *s* and *g* coordinates of the phasor plot for the decay $I(t)$ in each pixel were computed according to

$$g(\omega) = \frac{\sum_{i=1}^{n_t} I(t_i) \cos\left(\frac{2\pi t}{T}\right)}{\sum_{i=1}^{n_t} I(t_i)},$$

$$s(\omega) = \frac{\sum_{i=1}^{n_t} I(t_i) \sin\left(\frac{2\pi t}{T}\right)}{\sum_{i=1}^{n_t} I(t_i)} \tag{15}$$

where $T$ is the laser repetition period and $n_t$ the number of time points. For display and back gating, the phasor values were smoothed using a $5 \times 5$ uniform kernel. To generate the phasor plot, the phasor values for each pixel in an image were histogrammed into a $256 \times 256$ matrix with limits $0 \le g, s \le 1$. The histogram was weighted according to the intensity in each pixel.

## Hyperspectral unmixing of lifetime data

Using phasor analysis as described above, regions associated with liver cells, blood, collagen and the Src biosensor respectively were identified. For each region, the pixels were summed and a multispectral 'pattern' was generated by fitting the summed data in each spectral channel to a 4-component exponential model $P_{t,\lambda}(A, \tau)$.

$$P_{t,\lambda}(A, \tau) = g(t, \lambda) \otimes \sum_{i=1}^{4} A_{\lambda,i} \exp\left(-\frac{t}{\tau_{\lambda,i}}\right) \tag{16}$$

Where $g(t, \lambda)$ is the instrument response function, $A_{\lambda,i}$ is the contribution of the $i^{\text{th}}$ lifetime component $\tau_{\lambda,i}$ in channel $\lambda$. To identify the relative abundance of each component using these pre-determined patterns, non-negative least squares fitting was performed using each of the patterns for each pixel using the LAPACK routine 'nnls' (*Lawson and Hanson, 1995*) to determine the solution to the linear problem

$$
\begin{bmatrix}
P^{(1)}_{t=1,\lambda=1} & & P^{(m)}_{t=1,\lambda=1} \\
\vdots & & \vdots \\
P^{(1)}_{t=n_t,\lambda=1} & & P^{(m)}_{t=n_t,\lambda=1} \\
\vdots & \cdots & \vdots \\
P^{(1)}_{t=1,\lambda=n_\lambda} & & P^{(m)}_{t=1,\lambda=n_\lambda} \\
\vdots & & \vdots \\
P^{(1)}_{t=n_t,\lambda=n_\lambda} & & P^{(m)}_{t=n_t,\lambda=n_\lambda}
\end{bmatrix}
\begin{bmatrix}
\gamma_1 \\
\vdots \\
\gamma_m
\end{bmatrix}
=
\begin{bmatrix}
y_{t=1,\lambda=1} \\
\vdots \\
y_{t=n_t,\lambda=1} \\
\vdots \\
y_{t=1,\lambda=n_\lambda} \\
\vdots \\
y_{t=n_t,\lambda=n_\lambda}
\end{bmatrix}
\quad (17)
$$

where $y_{t=i,\lambda=j}$ is the intensity of the $i^{th}$ time point and $j^{th}$ channel and $\gamma_i \geq 0$ is the abundance of pattern $P^{(i)}_{t,\lambda}$ in the pixel.

**Video 6.** Tutorial screencast. Screencast documenting motion correction of FLIM data using *Galene*.
DOI: https://doi.org/10.7554/eLife.35800.023

## Rac1-GTP immunohistochemistry

The tissue was fixed in 10% buffered formalin solution overnight and embedded in paraffin using the swiss roll method (*Bialkowska et al., 2016*). Cut sections were de-paraffinised using xylene and rehydrated in graded ethanol washes. Antigen retrieval was performed in citrate buffer (S1699, pH = 6) for 30 min at 99°C and allowed to cool to RT for another 30 min. Endogenous peroxidase activity was subsequently quenched in 1.5% $H_2O_2$ before the application of 10% normal goat serum (NGS) in protein block (Dako) for 1 hr at RT. Slides were incubated overnight at 4°C with primary antibody (active Rac1-GTP, 1:400, NewEast Biosciences) in 10% NGS in protein block prior to applying secondary HRP-coupled anti-mouse antibody (Envision). Detection was performed with diamino-benzidine (DAB) for 5 min and slides counterstained with haematoxylin. Slides were digitalised at $20 \times$ magnification using a slide scanner (AperioCS2, Leica Biosystems). Data were analysed using QuPath (*Bankhead et al., 2017*). DAB and haematoxylin optical densities were computed for each pixel using colour deconvolution and regions containing crypts were manually identified. Nuclei within these regions were identified automatically using watershed cell detection based on the haematoxylin counterstain. Cell regions were then segmented by dilating the nuclear detections by 5 µm. The average DAB optical density was computed for each cell. These values were then averaged across each sample for n = 3 mice.

## Software availability

*Galene* is provided as an open source package with a graphical user interface and is available for download at https://galene.flimfit.org/ alongside user documentation, and is integrated directly into the *FLIMfit* analysis software (*Warren et al., 2013*). This software may be directly applied to data acquired on commercial microscope systems. The source code is available under the GPLv2 license at https://github.com/flimfit/Galene (copy archived at https://github.com/elifesciences-publications/Galene). The executables, manual and source code used in this manuscript are attached as Supplementary files. A tutorial screencast documenting the use of *Galene* is shown in *Video 6*.

## Acknowledgements

This study was supported by the National Health and Medical Research Council (NHMRC, project and fellowship funding), Cancer Institute NSW Early Career Fellowship, Cancer Council NSW, Cancer Australia, National Breast Cancer Foundation, St. Vincent's Clinical Foundation, Tour de Cure grants and a Len Ainsworth Pancreatic Cancer Research Fellowship. This project is made possible by an Avner Pancreatic Cancer Foundation Grant.

# Additional information

## Funding

| Funder | Grant reference number | Author |
|---|---|---|
| National Health and Medical Research Council | 1139865 | Sean C Warren<br>Max Nobis<br>Astrid Magenau<br>David Herrmann<br>Imogen Moran<br>Claire Vennin<br>James RW Conway<br>Pauline Mélénec<br>Thomas R Cox<br>Tri Giang Phan<br>Paul Timpson |
| Cancer Council NSW | RG 14-08 | Sean C Warren<br>Max Nobis<br>Astrid Magenau<br>David Herrmann<br>Claire Vennin<br>James RW Conway<br>Paul Timpson |
| Cancer Australia | | Sean C Warren<br>Max Nobis<br>Astrid Magenau<br>David Herrmann<br>Claire Vennin<br>James RW Conway<br>Paul Timpson |
| Tour de Cure, Australia | | Sean C Warren<br>Max Nobis<br>Astrid Magenau<br>David Herrmann<br>Claire Vennin<br>James RW Conway<br>Pauline Mélénec |
| Len Ainsworth Pancreatic Cancer Research Fellowship | | Sean C Warren<br>Max Nobis<br>Astrid Magenau<br>David Herrmann<br>Claire Vennin<br>James RW Conway<br>Paul Timpson |
| Avner Pancreatic Cancer Foundation | R3-PT | Sean C Warren<br>Thomas R Cox<br>Paul Timpson |
| National Health and Medical Research Council | 112468 | Sean C Warren<br>Max Nobis<br>Astrid Magenau<br>David Herrmann<br>Imogen Moran<br>Claire Vennin<br>James RW Conway<br>Pauline Mélénec<br>Thomas R Cox<br>Tri Giang Phan<br>Paul Timpson |

| National Health and Medical Research Council | 1089497 | Sean C Warren<br>Max Nobis<br>Astrid Magenau<br>David Herrmann<br>Imogen Moran<br>Claire Vennin<br>James RW Conway<br>Pauline Mélénec<br>Thomas R Cox<br>Tri Giang Phan<br>Paul Timpson |
|---|---|---|
| National Health and Medical Research Council | 1105640 | Sean C Warren<br>Max Nobis<br>Astrid Magenau<br>David Herrmann<br>Imogen Moran<br>Claire Vennin<br>James RW Conway<br>Pauline Mélénec<br>Thomas R Cox<br>Tri Giang Phan<br>Paul Timpson |
| National Health and Medical Research Council | 1129401 | Sean C Warren<br>Max Nobis<br>Astrid Magenau<br>David Herrmann<br>Imogen Moran<br>Claire Vennin<br>James RW Conway<br>Pauline Mélénec<br>Thomas R Cox<br>Tri Giang Phan<br>Paul Timpson |
| Cancer Institute NSW | 31329475 | Max Nobis<br>David Herrmann |
| Cancer Institute NSW | 31329983 | Max Nobis<br>David Herrmann |
| National Breast Cancer Foundation | IN-17-070 | David Herrmann<br>Paul Timpson |
| St. Vincent's Clinic Foundation | | David Herrmann<br>Paul Timpson |
| Biotechnology and Biological Sciences Research Council | Institute Strategic Programme Grant BB/P013384/1 | Heidi CE Welch |

The funders had no role in study design, data collection and interpretation, or the decision to submit the work for publication.

## Author contributions

Sean C Warren, Conceptualization, Software, Formal analysis, Investigation, Visualization, Writing—original draft; Max Nobis, David Herrmann, Claire Vennin, James RW Conway, Resources, Methodology, Writing—review and editing; Astrid Magenau, Formal analysis, Visualization, Writing—review and editing; Yousuf H Mohammed, Resources, Investigation, Writing—review and editing; Imogen Moran, Methodology, Writing—original draft; Pauline Mélénec, Investigation, Methodology, Writing—review and editing; Thomas R Cox, Methodology, Writing—review and editing; Yingxiao Wang, Jennifer P Morton, Heidi CE Welch, Douglas Strathdee, Kurt I Anderson, Resources, Writing—review and editing; Tri Giang Phan, Supervision, Funding acquisition, Methodology; Michael S Roberts, Resources, Funding acquisition, Writing—review and editing; Paul Timpson, Conceptualization, Supervision, Funding acquisition, Writing—review and editing

Author ORCIDs
Sean C Warren (iD) http://orcid.org/0000-0002-5253-7147
Max Nobis (iD) http://orcid.org/0000-0002-1861-1390
David Herrmann (iD) http://orcid.org/0000-0002-9514-7501
Thomas R Cox (iD) http://orcid.org/0000-0001-9294-1745
Yingxiao Wang (iD) http://orcid.org/0000-0003-0265-326X
Douglas Strathdee (iD) http://orcid.org/0000-0003-2959-4327
Kurt I Anderson (iD) http://orcid.org/0000-0002-9324-9598
Tri Giang Phan (iD) https://orcid.org/0000-0002-4909-2984

### Ethics

Human subjects: Experiments conducted on healthy human subjects using the DermaInspect were performed with informed consent and approval from the University of Queensland Human Research Ethics Committee (approval number 2007/197-2008001342). Experiments conducted on healthy human subjects using the hand-held multiphoton system were performed with informed consent and approval from Imperial College London (approval number 14IC2364).

Animal experimentation: All experiments were carried out in compliance with the Australian code for the care and use of animals for scientific purposes and in compliance with Garvan Institute of Medical Research/St. Vincent's Hospital Animal Ethics Committee guidelines (ARA 13/17, 16/13, 15/29).

### Decision letter and Author response

Decision letter https://doi.org/10.7554/eLife.35800.029
Author response https://doi.org/10.7554/eLife.35800.030

## Additional files

### Supplementary files

• Supplementary file 1. Acquisition and realignment parameters used through the study.
DOI: https://doi.org/10.7554/eLife.35800.024

• Transparent reporting form
DOI: https://doi.org/10.7554/eLife.35800.025

### Data availability

All data generated or analysed during this study are included in the manuscript and supporting files.

The following previously published dataset was used:

| Author(s) | Year | Dataset title | Dataset URL | Database, license, and accessibility information |
|---|---|---|---|---|
| Sherlock B, Warren SC, Alexandrov Y, Yu F, Stone J, Knight J, Neil MAA, Paterson C, French PMW, Dunsby C | 2018 | Data from: In vivo multiphoton microscopy using a handheld scanner with lateral and axial motion compensation | https://omero.bioinformatics.ic.ac.uk/omero/webclient/?show=project-4552 | Publicly available at OMERO (project number 4552) |

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
