## [Decision Letter]

Thank you for submitting your article "Galene: Removing physiological motion from intravital and clinical functional imaging data" for consideration by *eLife*. Your article has been reviewed by three peer reviewers, and the evaluation has been overseen by Anna Akhmanova as the Senior/Reviewing Editor. The reviewers have opted to remain anonymous.

The reviewers have discussed the reviews with one another and the Reviewing Editor has drafted this decision to help you prepare a revised submission.

Summary:

Motion artefacts represent recurrent challenges to sample subcellular-resolved image sequences by longitudinal intravital microscopy. Warren and coworkers present a motion correction algorithm which, quantitatively and over time, corrects for motion artefacts of fluorescence intensity features to improve the sampling of fluorescence lifetime analyses. By monitoring fluorescence intensity and established Rac and Src sensors in intestinal cypts, metastatic colorectal cancer cells during hepatic colonization and other models, they show remarkable gain in image quality over time, allowing to reconstruct cell and tissue topology with fine detail. This procedure was benchmarked against existing drift correction tools contained in popular broadly used software packages, including ImageJ (StagReg), Phyton (SIMA) and Imaris, with superior results obtained by their approach. The results for reasonably large scan fields for the chosen application are overall convincing, quantitatively benchmarked by autocorrelation and allow to extract a broad parameter space not readily available using alternative strategies, such as sampling smaller fields of interest, and detect even small changes of fluorescence lifetime.

Whereas the workflow is strong and the range of applications illustrates well the applicability of the drift correction across different organ types, sensors and quantities / types of drift, there are a number of concerns about the analysis and presentation that need to be addressed before the paper can be published.

Essential revisions:

1) The authors justify the requirement of image drift correction for acquiring more precise FLIM based data, and they measure this as the lifetime of fluorophores across what seems to be the entire imaging field. Since the lifetime is acquired by single photon counting, voxel per voxel with microsecond dwell-time, image drifts within the range corrected here should not be expected to alter this parameter significantly. Data supporting the need for image correction to obtain better FLIM measurements, compared to native images and images corrected using other drift corrections, are not provided. The authors should analyze whether the lifetime of the Rac or Src sensor differs between corrected and uncorrected images. Possibly, analyzing the lifetime in image subregions after defining ROIs at cellular and subcellular level will emerge as a selling point for their image correction, to e.g. extract differential responses in tissue niches, but so far no data are included to substantiate this idea.

2) In addition to single exponential decay analysis used here, the phasor approach is a well-established routine to extract lifetime information from multi-spectral datasets. As a particular strength, phasor analysis allows one to separate photon subsets within the same voxel. In addition, by back-gating, the phasor strategy can identify tissue subregions with defined lifetime properties, and this requires high image stability for identifying time-dependent processes. Thus, including data on superior performance of topographic analysis to improve phasor analysis will likely increase interest in applying this method of image correction. Furthermore, the authors are advised to test more complex double exponential fitting, because the single exponentials do not reflect the likely situation of different proportions of biosensors that do or do not exhibit FRET.

3) There is no detailed analysis of the effects of the pixel dwell time and frame size on the ability of Galene to work. Some mention of this can be found in the subsection “Algorithm 2. Simulation of TTTR data”, but no rigorous analysis is presented: for example: a smoothing kernel is used, but it is not clear how sensitive the overall method is to this. This would be essential for a technical paper that could then be followed by others in the field, especially as there is even some advice on these matters on the http://galene.readthedocs.io website and in the user manual. For example, it be would important to see analysis using real data (not simulated) of how the ability of Galene to function drops off as the scan rate gets slower. Similarly, real examples should be shown where the scan axis is varied to align with the breathing motion or not on the same sample.

4) There is ambiguity about the level of advance on the methodology of Lucas-Kanade. It appears that the basis of the algorithms is the same, but that the code is greatly improved and able to run using GPU systems, which makes it much quicker. There is also the highly significant advance that the code can handle the time tagged time resolved TCSPC data. The Galene website shows some of the same data as in the manuscript as a screen shot, but worryingly the 'before' image seems transposed between the website and Figure 4 of the manuscript. This is probably a simple cut and paste error, but should be corrected. A more technical presentation might help to clarify the novelty.

5) Writing and presentation: The paper is written half as a technical paper and half as a primary research paper with various intriguing highlights of data, such as modulation of Src or Rac activity in vivo. Unfortunately, the technical part lacks the detail that will be key for other users to benefit from Galene and the research part lacks any consistent narrative. We strongly recommend re-focusing on a more technical paper with only one example of data from each type of imaging device.

Furthermore, the results are presented model by model, however the actual content per image is partly redundant, given that the overall suitability for image stabilization is clear by Figure 4. As consequence, the text is quite long and meandering, moving from model to model. Because these imaging windows have been described elsewhere, the authors might want to consider focusing on conceptual advances, e.g. moving from whole-field analysis to subregion analysis, from single-exponential decay analysis to multispectral analysis (phasor gating), and thereby progress from dataset to dataset. The findings on improving intensity data could be strongly condensed. Consequently, several largely iterative datasets currently presented as main figures might be well-suited for supplementary documentation, including most of Figure 3, Figure 5 and 8 (both lack FLIM data; intensity-based corrections are also shown in Figure 2, 4, 6, 9).

6) The reported effects of scopolamine are interesting and potentially very important. However, to make strong conclusions, the authors must corroborate them using an orthogonal biochemical method, or the conclusions need to be softened.

7) It is completely unclear what has been done to derive the diffusion metrics in the Figure 9. Clearly the authors are not measuring diffusion in the normal way, which is usually measured as area/time. This is a further example of the inadequate presentation. Also, E-cadherin diffusion would be in the plane of the membrane, which is perpendicular to the image shown. Transport in and out of the membrane compartment will not be a diffusive process for an integral membrane protein. The authors are advised to remove this part.

---

## [Author Response]

Essential revisions:1) The authors justify the requirement of image drift correction for acquiring more precise FLIM based data, and they measure this as the lifetime of fluorophores across what seems to be the entire imaging field. Since the lifetime is acquired by single photon counting, voxel per voxel with microsecond dwell-time, image drifts within the range corrected here should not be expected to alter this parameter significantly. Data supporting the need for image correction to obtain better FLIM measurements, compared to native images and images corrected using other drift corrections, are not provided. The authors should analyze whether the lifetime of the Rac or Src sensor differs between corrected and uncorrected images.

The reviewers are correct that the arrival time of each photon event is not affected by motion correction. However, motion during image acquisition can effectively convolve pixels together from different regions, for example pixels from the biosensor and tissue autofluorescence. Since the difference in lifetime between the biosensor and autofluorescence is often significantly larger than the expected changes in the biosensor lifetime, this can introduce a significant artefact into the data. To demonstrate this effect quantitatively, we have now compared the Src biosensor lifetime in pre- and post-correction FLIM images (Figure 4G, H). We observe a significant difference in the biosensor lifetime before and after correction and have added the following text:

“To evaluate the impact of motion correction on our ability to quantify Src activity in dataset, we analysed the lifetime of the uncorrected data in the same way (Figure 5H). […] This increased variability abolishes our ability to statistically distinguish the conditions, highlighting the importance of motion correction to obtain robust results in this context (compare in Figure 5G to H).”

Possibly, analyzing the lifetime in image subregions after defining ROIs at cellular and subcellular level will emerge as a selling point for their image correction, to e.g. extract differential responses in tissue niches, but so far no data are included to substantiate this idea.

We agree with the reviewer; to demonstrate this point, we have performed a subcellular quantification of the Rac1 intestinal crypt data (Figure 4F), adding the following text:

“Without motion correction, it is extremely difficult to identify subcellular compartments in a majority of the intestinal crypt data. […] We observed a lower level of Rac1 activation the basal membrane compared to the apical membrane after application of PMA or scoloplamine, suggesting a potential negative regulation of Rac1 at the basal membrane.”

We believe that, together, this analysis provides a compelling argument for the use of motion correction in FLIM data.

2) In addition to single exponential decay analysis used here, the phasor approach is a well-established routine to extract lifetime information from multi-spectral datasets. As a particular strength, phasor analysis allows one to separate photon subsets within the same voxel. In addition, by back-gating, the phasor strategy can identify tissue subregions with defined lifetime properties, and this requires high image stability for identifying time-dependent processes. Thus, including data on superior performance of topographic analysis to improve phasor analysis will likely increase interest in applying this method of image correction.

To highlight the ability to perform phasor analysis on the motion corrected data, we have now demonstrated the use of phasor analysis with both the Rac1 and Src FRET data. Using the Rac1 FRET data, we showed the ability of phasor analysis to distinguish between the biosensor and tissue autofluorescence (new Figure 4C).

“To quantify the data, we first used phasor analysis (Figure 4C, phasor analysis of image shown in Figure 4B) to separate the biosensor fluorescence (blue gate) from the tissue autofluorescence (red gate).

For the Src FRET data we now show the ability of phasor analysis to distinguish between the biosensor, vasculature, collagen network and hepatocytes using phasor gating (new Figure 5D) and added the following text:

“Figure 5Di shows a merged intensity image in the three spectral channels used and Figure 5Dii and iii show the temporal phasor of the 525/50 channel and the spectral phasor respectively. Phasor gates associated with Src-FRET, hepatocytes, the vasculature and collagen were identified as shown and used to identify the associated region in the image (Figure 5Div).”

In the Discussion, we added:

“We have demonstrated that it is possible to robustly analyse motion corrected FLIM data using a number of methods which require a high signal noise levels and sample stability. We used reconvolution based fitting to a mono-exponential or complex donor FRET model to extract quantitative parameters about FRET activity, phasor analysis to perform model-free topographic analysis of exogenous and autofluorescent components and hyperspectral analysis to determine the abundance of different autofluorescent signals in the liver.”

We have added a corresponding section to the Materials and methods.

Furthermore, the authors are advised to test more complex double exponential fitting, because the single exponentials do not reflect the likely situation of different proportions of biosensors that do or do not exhibit FRET.

We agree that the use of a more complex fitting model is valuable. Since the Rac1 biosensor uses ECFP as a donor, which has a complex decay profile even in the absence of FRET, we analysed the intestinal crypt data using a complex-donor FRET model (Warren et al., 2013), instead of a double exponential model, to determine the proportion of biosensors that exhibit FRET (throughout Figure 4). We have added the following text and an appropriate additional section in the Materials and methods.

“The Rac1 biosensor contains an ECFP donor which has a complex decay profile, dominated by contributions from two conformations with similar spectral profiles. […] By fitting the contributions of each component, we can estimate the fraction of active Rac1 biosensor in each pixel as shown in Figure 4A.”

3) There is no detailed analysis of the effects of the pixel dwell time and frame size on the ability of Galene to work. Some mention of this can be found in the subsection “Algorithm 2. Simulation of TTTR data”, but no rigorous analysis is presented: for example: a smoothing kernel is used, but is not clear how sensitive the overall method is to this. This would be essential for a technical paper that could then be followed by others in the field, especially as there is even some advice on these matters on the http://galene.readthedocs.io website and in the user manual. For example, it be would important to see analysis using real data (not simulated) of how the ability of Galene to function drops off as the scan rate gets slower. Similarly, real examples should be shown where the scan axis is varied to align with the breathing motion or not on the same sample.

We thank the reviewers for this suggestion. We have now added a detailed discussion of the effect of the scan rate (or, equivalently pixel dwell time) and the relative alignment of the scan axis and the sample motion using real data acquired in the pancreas in the new section “Evaluation of the effect of image scan configuration on motion correction performance” and new Figure 3. We have explored the effect of the number of realignment points and the smoothing kernel used in new section “Evaluation of the effect of realignment parameters on motion correction performance” and new Figure 3—figure supplement 1. For brevity, these sections are not quoted here. Additionally, we have clarified the use of the Gaussian smoothing filter in the section “Reconstruction of frames from TTTR data”.

“To reduce noise, an elliptical Gaussian filter with a user-controllable radius in the fast scan axis and a 1px radius in the slow scan axis is applied. This filter is applied to the reference frame and each frame used for realignment but not to the final, realigned data.”

4) There is ambiguity about the level of advance on the methodology of Lucas-Kanade. It appears that the basis of the algorithms is the same, but that the code is greatly improved and able to run using GPU systems, which makes it much quicker. There is also the highly significant advance that the code can handle the time tagged time resolved TCSPC data.

We apologise for the lack of clarity in presentation here. The reviewer is correct in their assessment. The Lucas-Kanade framework is a more general approach for estimating image warps. We are applying this algorithm to data acquired using a raster pattern. We have now significantly expanded the exposition of the method in the new section “Correction for motion in time-tagged, time-resolved FLIM data” (expanded from a previous discussion in the Introduction, not quoted here for brevity) which makes this clear. We also discuss the novel extensions presented here in the Discussion:

“Galene uses a fitting approach that explicitly accounting for the raster scan pattern performed by laser scanning microscopes to determine sample motion both between frames and during each frame. […] These modifications reduce the time required to perform motion correction by a factor of 30 for typical volumetric datasets and open the door to online motion correction in the future.”

The Galene website shows some of the same data as in the manuscript as a screen shot, but worryingly the 'before' image seems transposed between the website and Figure 4 of the manuscript. This is probably a simple cut and paste error, but should be corrected. A more technical presentation might help to clarify the novelty.

We apologise for this oversight. In an earlier version of the (non-motion correction) FLIM reconstruction software, we did not correct for the difference in row-major and column-major interpretations of matrix data between C++ and Matlab, leading to a transposition error when loading the FLIM data for analysis. This was corrected; however we had not reflected this change in the figure. We have now corrected the figure.

5) Writing and presentation: The paper is written half as a technical paper and half as a primary research paper with various intriguing highlights of data, such as modulation of Src or Rac activity in vivo. Unfortunately, the technical part lacks the detail that will be key for other users to benefit from Galene and the research part lacks any consistent narrative. We strongly recommend re-focusing on a more technical paper with only one example of data from each type of imaging device.Furthermore, the results are presented model by model, however the actual content per image is partly redundant, given that the overall suitability for image stabilization is clear by Figure 4. As consequence, the text is quite long and meandering, moving from model to model. Because these imaging windows have been described elsewhere, the authors might want to consider focusing on conceptual advances, e.g. moving from whole-field analysis to subregion analysis, from single-exponential decay analysis to multispectral analysis (phasor gating), and thereby progress from dataset to dataset. The findings on improving intensity data could be strongly condensed. Consequently, several largely iterative datasets currently presented as main figures might be well-suited for supplementary documentation, including most of Figure 3, Figure 5 and 8 (both lack FLIM data; intensity-based corrections are also shown in Figure 2, 4, 6, 9).

We have now refocussed the manuscript on discussion of the technical aspects of motion correction, using the data where appropriate to illustrate these points. As discussed in the response to points 3 and 4, we have significantly expanded the description of the realignment process and added new technical sections discussing the effect of the scan parameters and realignment options on the correction performance to help the reader make the most effective use of this technology. In response to point 2, we have refocussed the text to further discuss different analysis techniques that may be applied to motion-corrected data.

Specifically, we have refocussed the intravital Rac1 FRET data so that the pancreas data is used to discuss the effect of scan speed and angle relative to motion and the realignment parameters. We now use the intestinal crypt data as an example of analysis using the complex-donor FRET method and to demonstrate the ability to detect and reject frames where the sample has moved out of frame. We have significantly reworked and simplified the ex-vivo intestinal crypt data. We now simply highlight the difference between pharmacological motion reduction with Scopolamine and the present motion correction approach. Instead, we focus on the ability to perform sub-region analysis using this dataset.

In line with this, we have removed the previous text and figure relating to motion correction of Rac1-FRET data in the mammary gland and pancreas which was largely iterative. As requested, we then moved the benchmarking (previous Figures 3 and 5) to the supplement. Further, we have significantly reduced the duplication in the figures, for example removing the displacement traces in Figures 4B and 5.

We have also reduced the background to the Src intrasplenic experiment to the minimum required to understand the experiment and its motivation. We then refocussed this section to demonstrate phasor analysis and backgating and the effect of motion correction on FLIM quantification due to background fluorescence, adding the quantitative comparison with uncorrected FLIM data.

We have removed the murine liver NADH window data, focussing the NADH autofluorescence data on the ability to correct for motion in 3D in clinical imaging data using either online axial motion correction (Figure 5A) or full 3D motion correction using *Galene* (Figure 5B).

Finally, we have removed the ICS data as requested, leaving the lymph node as the sole non-FLIM dataset, which highlights the ability to apply raster-pattern aware correction to 3D data. We note that, to our knowledge, this is the first demonstration of this ability even in intensity data. We have condensed the discussion of this data.

These changes occur systemically throughout the manuscript so are not quoted here.

6) The reported effects of scopolamine are interesting and potentially very important. However, to make strong conclusions, the authors must corroborate them using an orthogonal biochemical method, or the conclusions need to be softened.

We agree with the reviewers, to address this point we have performed immunhistochemical staining against active Rac1-GTP in freshly excised tissue to confirm that Rac1 activity is increased in intestinal crypts following treatment with Scopolamine (new Figure 4—figure supplement 1).

7) It is completely unclear what has been done to derive the diffusion metrics in the Figure 9. Clearly the authors are not measuring diffusion in the normal way, which is usually measured as area/time. This is a further example of the inadequate presentation. Also, E-cadherin diffusion would be in the plane of the membrane, which is perpendicular to the image shown. Transport in and out of the membrane compartment will not be a diffusive process for an integral membrane protein. The authors are advised to remove this part.

We apologise for the unclear presentation and have removed this section.